# Prediction of the Hydrodynamic Forces for a Ship Oscillating in Calm Water by an Improved Higher Order Rankine Panel Method

**Chun-Hsien Wu * and Ming-Chung Fang**

Department of Systems and Naval Mechatronic Engineering, National Cheng Kung University, Tainan 70101, Taiwan
* Correspondence: wuchunhsien75@gmail.com

**Abstract:** This paper presents a frequency-domain Rankine source method based on a biquadratic B-spline scheme with an improved radiation mechanism. The improved radiation mechanism, based on the introduction of spatially varying Rayleigh artificial damping in addition to the simplified Seto's radiation boundary conditions, is considered for modeling radiation of generated waves at various $\tau$ conditions, where $\tau = \omega U / g$ including the undercritical condition ($\tau < 0.25$); this condition is present when a ship undergoes slow translation or low oscillatory frequency. In evaluations, the proposed method yields accurate solutions for unsteady flows produced by an oscillating, translationally moving submerged singularity. The radiation problem induced by a RIOS bulker is solved to have the resultant added mass and damping coefficients for further comparisons with the experimental data and the public numerical prediction by a simplified combined method at a wide $\tau$ region.

**Keywords:** Rankine source method; radiation boundary condition; Rayleigh artificial damping; radiation problem; hydrodynamic coefficients

## 1. Introduction

Green function and Rankine source methods are commonly used to solve flow problems in marine engineering. The advantages of the Green function method are twofold: first, the boundary integral equation is constructed from only the wetted body surface, and second, the free surface boundary condition and the radiation boundary condition at infinity are automatically satisfied. The research studies in [1–3] have successfully applied the Green function method to seakeeping problems. However, a converged and efficient numerical way to evaluate the Green function singularity and the proper numerical treatment of the waterline integral are critical to Green function method; meanwhile they are thought of as limitations in usage of Green function implementation. To address these limitations, the Rankine source method, where the Rankine source is used as the singularity, is considered. The Rankine source method facilitates effective evaluation of the employed singularity and there is no need for treatment of the waterline integral; this method is thus thought to be promising in practical applications.

The pioneer work for use of the Rankine source method is from Dawson [4] for predicting the resistance of a body sailing in calm water. Dawson's success in solving this steady flow problem began the prevalent use of the Rankine source method for solving unsteady flow problems. Bai and Yeung [5] are the pioneers credited for application of the frequency-domain Rankine source method in solving unsteady flow problems, and achieving good prediction of two axi-symmetric three-dimensional bodies, such as sphere and ellipsoid, with the existing results. As illustrated by Bai and Yeung [5], in addition to the infinite-depth case, the Rankine source method can be applied in the finite-depth case by constructing the relevant boundary equation with the additional Rankine singularity

distribution over the bottom and taking the finite-depth dispersion relation into account for determining the corresponding wave number and wave length at constant finite depth.

Since then, frequency-domain analyses of the realistic ship's unsteady flow problems using the Rankine source method have been reported by Nakos [6], Bertram [7], Yasukawa [8], Iwashita and Ito [9], Yang et al. [10], Söding [11,12], Lyu and Moctar [13], Kim and Kim [14] and others. Furthermore, the time-domain Rankine source method using the linear, weakly nonlinear or weak scatter assumption on free surface boundary conditions are considered by Kring [15], Kim and Kim [16], Huang [17], Singh and Sen [18] and Datta et al. [19], for attempts to deal with the flow problem induced by the ship in large amplitude motion. To handle the full nonlinearity existing in interaction between the body and water, the method of mixed Eulerian-Lagrangian, in which the evolution in the dynamic and kinematic free surface boundary conditions are considered, is introduced by Longuet-Higgins and Cokelet [20] to simulate the two-dimensional wave profile variation in time. Its succeeding application in the three-dimensional problems are investigated by Cao [21], Beck [22,23], Kashiwagi [24] and Abbasnia [25]. However, even though these variants of the time-domain Rankine source methods are proposed and have progressed in past decades, the frequency-domain method is considered to be the most mature and efficient way of dealing with various flow conditions in engineering practices. For this reason, the present paper aims to use the frequency-domain Rankine source method for wider applications in the marine field.

To satisfy the free surface boundary condition, the computation domain required for constructing the boundary integral equation covers the body surface and the extent of the free surface. Although the allocation of the free surface implies more computational cost, it provides good flexibility in satisfying the variant of the free surface boundary condition. However, the accompanying problems of wave distortion due to free surface discretization and avoidance of wave reflection must be addressed. Jensen et al. [26] and Sclavounos [27] have improved the method by reducing numerical dispersion and dissipation. To ensure that the radiation boundary condition is satisfied on the free surface, Dawson [4] employed the upwind differential, Raven [28] analyzed the source shift, Sclavounos [27] applied the rigid-lid condition and Yasukawa [8] suggested introduction of the Rayleigh artificial damping term. However, these numerical studies focused on solving the flow problem under overcritical conditions, in which $\tau(=\omega U/g) > 0.25$ and all the generated wave systems propagate backward; their proposals may not be appropriate for the flow problem under undercritical conditions, where $\tau < 0.25$ and part of the generated wave system scatters ahead of the translating vessel. Herein, $\tau$ is the oscillatory frequency of the vessel, $U$ is the forward speed, and $\tau$ is thought of as an index for interpreting the generated wave system.

Das and Cheung [29] and Yuan et al. [30] have attempted to address the aforementioned limitation by formulating an extended Sommerfeld radiation boundary condition derived from the forward-speed-induced Doppler shift in circular waves at an additional control surface. In a recent study by Iwashita et al. [31,32], the joint conditions for describing wave radiation are proposed to be blended with the Rankine source method. These joint conditions, derived based on the concept of a combined domain, were employed at the control surfaces and defined as the induced velocity potential and its normal derivative simulated by the allocation of Green function singularity. Furthermore, the joint condition is suggested as the resultant ratios of the normal derivatives to velocity potential, and such a technique, which is called the simplified combined method, is used for seakeeping prediction of ship, e.g., the studies in [32,33]. However, implementation of the extended Sommerfeld radiation or joint conditions requires good control surface allocation and increases the number of unknowns and computational cost.

In order to have the continuous description of flow and pressure fields, the velocity potential modeled in spline form is considered in constructing the boundary integral equation; this lies at the heart of the so-called higher-order method. Hsin et al. [34] constructed a model using B-splines for a two-dimensional flow problem. Scholars such as Nakos [6],

Maniar [35], Coaxley [36], and Kim and Kim [37] have extended this insight to three-dimensional steady and unsteady flow problems. Furthermore, Nakos and Sclavounos [38] discuss the stability analysis, which is based on the dispersion relation in continuous and discretized problems, for the B-spline scheme and propose the stability criterion.

The present study presents a frequency-domain Rankine source method that is based on a B-spline scheme. This method can be used to solve unsteady flow problems regardless of the $\tau$ condition (making this method practical for use). This method contrasts with the unsteady Rankine source approaches in the literature that are applicable only to the overcritical condition. Specifically, the present study constructs the resultant boundary integral equation by employing simplified Seto's explicit conditions and Rayleigh artificial damping to accommodate wave radiation. In evaluation of its effectiveness, the proposed method was successfully used to solve for the flow field generated by a single submerged disturbance source under undercritical and overcritical conditions [39]. The present bi-quadratic B-spline scheme is taken into account for predicting the hydrodynamic forces and moments induced by forced motion of a realistic bulk carrier; its predicted coupled added mass and damping coefficients are compared with the experimental data and the numerical result of the simplified combined method to investigate applicability of the present method in a broader region of $\tau$.

## 2. Mathematical Formulation

### 2.1. Boundary Value Problem

Figure 1 illustrates the boundary problem of an oscillating vessel translating at steady forward speed $U$ in deep calm water. The oscillatory frequency is represented as $\omega$. The adopted right-handed Cartesian coordinate frame $o - xyz$ undergoes translational motion with the vessel on the positive $x$-axis, pointing to bow; its positive $z$-axis points upward with the frame origin $o$ located on the gravity center of the vessel at still water. As presented in Figure 1, the finite computation extents conducted for the Rankine source panel scheme includes the free surface and wetted surface of the vessel, and herein are denoted by $\Sigma_F$ and $\Sigma_H$, respectively.

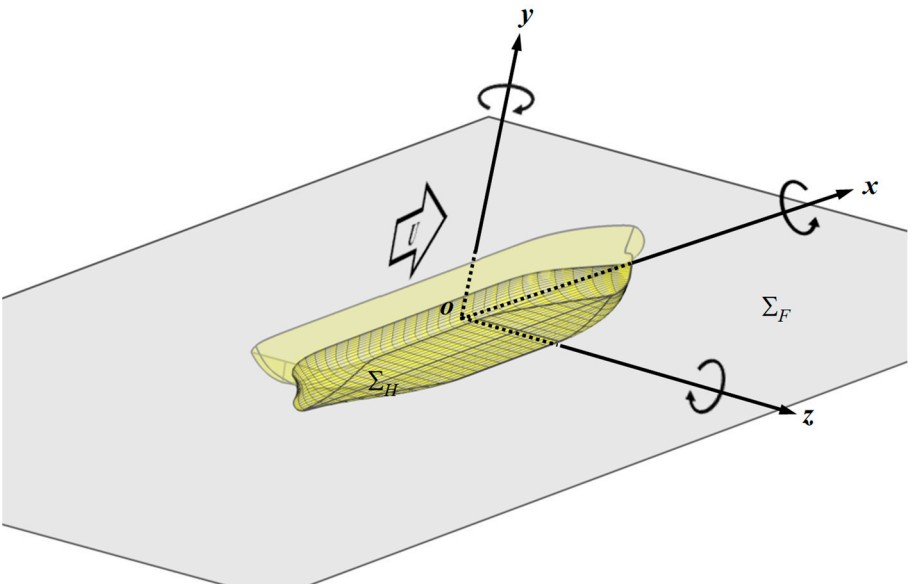

**Figure 1.** Coordinate system.

The unsteady motion response of the vessel in calm water and the associated ambient flow are considered to be periodical with the ship oscillating frequency given by $\omega$. The forced motion responses of the vessel are described in the six degrees of freedom oscillatory amplitudes of the vessel in calm water and are denoted by $\xi_j (j = 1{\sim}6)$, which stands for surge, sway, heave, roll, pitch and yaw movement, respectively.

The fluid throughout the domain in deep water, in which the Laplace equation is satisfied, is inviscid, incompressible flow and irrotational. Thus, the flow field can be described in the framework of velocity potential in the present study. Let $t$ denote time and $\vec{p} = (x, y, z)$ means the position vector. The total induced flow potential $\Psi\left(\vec{x}, t\right)$ by the translating vessel in calm water is expressed as

$$\Psi\left(\vec{p}, t\right) = U\left[\Phi\left(\vec{p}\right) + \phi\left(\vec{p}\right)\right] + \varphi\left(\vec{p}\right)e^{i\omega t} \tag{1}$$

where $i = \sqrt{-1}$; $\Phi$ is the double-body flow and is considered as the basis flow resulting from the presence of the translating vessel in rigid water plane. The velocity potential $\phi$ and $\varphi$ stand for the steady and unsteady flow. Furthermore, the radiation flow component of unsteady potential flow reads as follows

$$\varphi\left(\vec{p}\right) = \sum_{j=1}^{6} \xi_j \varphi_j\left(\vec{p}\right) \tag{2}$$

in which $\varphi_j (j = 1\sim6)$ stands for the velocity potential per unit $j$-mode motion displacement.

In direct methodology of the Rankine source scheme, the unsteady velocity potential in fluid domain is expressed in terms of a normal dipole distribution of moment $\varphi$ and a source distribution of strength $\varphi/\partial n$ distributed over the boundary surfaces of $\Sigma_F$ and $\Sigma_H$, with the Rankine source as the kernel singularity, as in the following boundary integral equation:

$$2\pi\varphi\left(\vec{p}\right) = \iint\limits_{\Sigma_F + \Sigma_H} \left[G\left(\vec{p}, \vec{q}\right)\frac{\partial\varphi}{\partial n}\left(\vec{q}\right) - \frac{\partial G}{\partial n}\left(\vec{p}, \vec{q}\right)\varphi\left(\vec{q}\right)\right]ds \tag{3}$$

where $\varphi$ stands for either one of $\varphi_j$, $(j = 1\sim6)$, $\vec{p} = (x, y, z)$ and $\vec{q} = (x', y', z')$ represents the field point in the fluid and the source point over the boundary surfaces, respectively. $G\left(\vec{p}, \vec{q}\right)$ represents the induced velocity potential at point $\vec{p}$ by the unit strength Rankine source at point $\vec{q}$ and equals $1/r$; in which $r = \sqrt{(x - x')^2 + (y - y')^2 + (z - z')^2}$. In the present study, the velocity potential is generally approximated in the B-spline form as $\varphi\left(\vec{q}\right) \approx \sum_{m=1}^{M} B_m^{(2,2)}\left(\vec{q}\right)\sigma_m$; in which $\sigma_m$ denotes the spline coefficients of $M$ control units corresponding to the basis function $B_m^{(2,2)}$ that is a product of two univariate quadratic polynomials described in more detail in the later section. Therefore, the spline coefficients $\sigma_m$ are the unknowns in the present solving of the boundary value problem.

A series of boundary integral equations raised at various field points $\vec{p}$ collocated on, and end conditions around, the boundary surfaces $\Sigma_F$ and $\Sigma_H$ are collected to construct a linear system of simultaneous equations for solving the spline coefficients for approximating velocity potential over the boundary surfaces. Once the solution of spline coefficients is obtained, we can have the flow potential, the spatial derivatives and the hydrodynamic forces acting on the vessel accordingly.

*2.2. Boundary Conditions*

To completely construct the unsteady wave flow problem, the flow disturbance due to the steady translation of the body is considered in the derivations of relevant boundary conditions and should be solved prior to solving the unsteady flow problem.

As mentioned in the preceding section, the velocity potentials $\Phi$ and $\phi$ denote the double-body flow and the steady flow, which are induced due to the presence of the body on free surface and the steady translation. Normally, the steady potential is disregarded in solving an unsteady flow problem because the double-body flow is considered to be the basis flow and is the dominant term. Hence, the relevant velocity vector due to the body steady translation is written as by $\nabla\Phi$, normalized by forward speed $U$. The phenomenon

of no flow flux penetrating through the boundary surface gives the boundary conditions for the double-body flow as:

$$\frac{\partial \Phi}{\partial z} = 0 \; z = 0 \tag{4}$$

$$\frac{\partial \Phi}{\partial n} = n_1 \in S_H \tag{5}$$

where $n_1$ denotes the $x$-component of normal vector on the hull.

On the exact free surface, the fluid flow is required to satisfy kinematic and dynamic boundary conditions—that is, the normal velocities of the fluid and of the boundary surface must be equal, and the pressure on the free surface must be equal to atmosphere. The single form of the boundary condition can be given by taking a substantial derivative of the dynamic boundary condition and combining the kinematic boundary condition to eliminate the wave elevation term. It is further linearized with respect to the basis double-body flow by neglecting the higher order terms and with Taylor series expanding at the still water plane z = 0, we can have resultant linearized free surface boundary condition for unsteady flow as follows

$$\frac{\partial \varphi}{\partial n}\left(\vec{q}\right) = \left\{ \begin{array}{c} \kappa - i\tau\left(\Phi_{xx} + \Phi_{yy}\right) \\ -\left[2i\tau\Phi_x + \frac{2}{\kappa_0}\left(\Phi_x\Phi_{xx} + \Phi_y\Phi_{xy}\right) + \frac{1}{\kappa_0}\Phi_x\left(\Phi_{xx} + \Phi_{yy}\right)\right]\frac{\partial}{\partial x} \\ -\left[2i\tau\Phi_y + \frac{2}{\kappa_0}\left(\Phi_x\Phi_{xy} + \Phi_y\Phi_{yy}\right) + \frac{1}{\kappa_0}\Phi_y\left(\Phi_{xx} + \Phi_{yy}\right)\right]\frac{\partial}{\partial y} \\ -\frac{1}{\kappa_0}\left(\Phi_x\frac{\partial}{\partial x} + \Phi_y\frac{\partial}{\partial y}\right)^2 \end{array} \right\} \varphi\left(\vec{q}\right) \quad z = 0 \tag{6}$$

where the parameters shown in Equation (6) are defined as $\kappa = \frac{\omega^2}{g}$, $\kappa_0 = \frac{g}{U^2}$ and $\tau = \frac{\omega U}{g}$.

With regard to the boundary condition at the body surface, the kinematic condition, in which the normal component of the fluid velocity is equal to the component of body velocity, is considered in aspect of the space-fixed coordinate. It is further considered that the unsteady force due to the oscillatory movement of the body is balanced by the unsteady flow. The linearization manipulation is taken by the Taylor series expanding with respect to the equilibrium position of the body and dropping higher order terms, and the linearized body boundary condition for the radiation problem is expressed in the form

$$\frac{\partial \varphi_j}{\partial n} = i\omega n_j + m_j j = 1 \sim 6 \in S_H \tag{7}$$

in which $(n_1, n_2, n_3) = \vec{n}$, $(n_4, n_5, n_6) = \vec{x} \times \vec{n}$; $\vec{n}$ denotes the normal vector and points inwards of the hull. The so-called $m$-term represents the disturbance effects due to the steady flow in unsteady flow problem, and its components are defined by

$$m_j = \begin{cases} -\left(\vec{n} \cdot \nabla\right)\nabla\Phi & j = 1, 2, 3 \\ -\left(\vec{n} \cdot \nabla\right)\left(\vec{x} \times \nabla\Phi\right) & j = 4, 5, 6 \end{cases} \in S_H \tag{8}$$

*2.3. Hydrodynamic Coefficients*

Once the unknown double-body flow and the unsteady flow potentials, due to the oscillatory motion, are solved, the resultant wave elevation on free surface and the pressure on the vessel can be evaluated. The wave elevation and the resultant linear hydrodynamic pressure can be expressed as Equations (9) and (10), respectively.

$$\zeta(x,y) = -\frac{1}{g}(i\omega + \nabla\Phi \cdot \nabla)\varphi \; z = 0 \tag{9}$$

$$p\left(\vec{p}\right) = -\rho(i\omega + \nabla\Phi \cdot \nabla)\varphi \in S_H \tag{10}$$

in which $\rho$ is the fluid density. The pressure integration over the mean wetted surface of the vessel gives the hydrodynamic forces and moments, which can be further classified as the radiation force and wave excitation force. The radiation force, associated with the monochromic oscillatory motions of vessel in six degrees of freedom, is derived from the radiation potential as

$$D_i = -\rho \sum_{j=1}^{6} \xi_j \left[ \iint_{\Sigma_H} (i\omega + \nabla\Phi \cdot \nabla)\varphi_j n_i ds \right] = \sum_{j=1}^{6} \xi_j \left( \omega^2 a_{ij} - i\omega b_{ij} \right) \ i = 1 \sim 6 \quad (11)$$

where the terms $a_{ij}$ and $b_{ij}$ represent the added mass and damping coefficients, respectively (extracted from the real and image part in $D_i$). Here the subscript $ij$ indicates that the coupled coefficient of added mass or damping acting at the $i^{\text{th}}$ degree of freedom is induced by the forced motion of the $j^{\text{th}}$ mode. Furthermore, $a_{ij}$ and $b_{ij}$ can be written in the form of the integral as Equations (12) and (13), respectively.

$$a_{ij} = -\frac{\rho}{\omega^2} \mathbb{R} \left\{ \iint_{\Sigma_H} (i\omega\varphi_j + \nabla\Phi \cdot \nabla\varphi_j) n_i ds \right\} \in S_H \quad (12)$$

$$b_{ij} = \frac{\rho}{\omega} \mathbb{F} \left\{ \iint_{\Sigma_H} (i\omega\varphi_j + \nabla\Phi \cdot \nabla\varphi_j) n_i ds \right\} \in S_H \quad (13)$$

*2.4. Radiation Boundary Condition*

The present study is conducted based on the spline Rankine source scheme with collocation method, and the resultant linear algebraic system is an underdetermined system since the equations raised on one collocation point per panel are insufficient to adequately solve the unknowns of spline coefficients. Thus, the additional end conditions are required to make the system determined and be solvable.

Among end conditions imposed around the extents of the free surface and vessel, the upstream truncation of free surface extent is considered of most dominant and it should be treated carefully. As illustrated in Figure 2, the generated wave systems by the forward-moving vessel at a two-dimensional aspect can be classified into two major wave groups, which are referred to as the A-wave and B-wave, respectively. A-wave is generated behind the moving vessel and propagates downward, which is grouped by the shorter $a_1$-wave and the longer $a_2$-wave. By contrast, B-wave propagates at the forward direction, which consists of $b_1$-wave and $b_2$-wave at different group velocities.

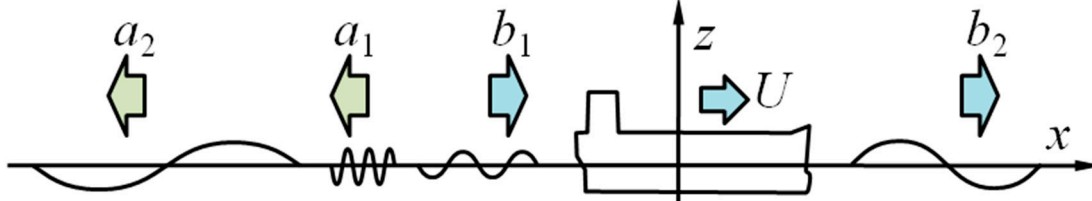

**Figure 2.** Wave systems for a forward-moving vessel.

The associated group velocities can be determined by the following equations.

$$\left.\begin{array}{c}C_g^{a_1}\\C_g^{a_2}\end{array}\right\}=\frac{\sqrt{2}U}{2\sqrt{\left(1+2\tau\pm\sqrt{1+4\tau}\right)}}\tag{14}$$

$$\left.\begin{array}{c}C_g^{b_1}\\C_g^{b_2}\end{array}\right\}=\frac{\sqrt{2}U}{2\sqrt{\left(1-2\tau\pm\sqrt{1-4\tau}\right)}}\tag{15}$$

It is worth noting that $C_g^{b_2}>U$ and concludes that the $b_2$-wave can propagate ahead of the vessel only when undercritical condition $\tau<0.25$. Therefore, this significant phenomenon of the $b_2$-wave leads to demand for appropriate radiation boundary conditions as the end conditions upstream for both overcritical and undercritical conditions.

Based on the authors' previous study [39], the explicit radiation boundary conditions set, derived from the conditions proposed by Seto [40], at upstream truncation are found. However, to avoid the overdetermined system resulted from the triple conditions for undercritical condition in numerical implementation, a common formula is simplified to be as follows,

$$\left(\frac{\partial}{\partial x}+ik_2\cos\delta\right)\left(\frac{\partial}{\partial x}-i\kappa_0\tau\right)\varphi\sim0\tag{16}$$

$$\left(\frac{\partial}{\partial x}+ik_2\cos\delta\right)\left(\frac{\partial}{\partial x}-i\kappa_0\tau\right)^2\varphi\sim0\tag{17}$$

In Equations (16) and (17), $k_2$ and $\delta$ stand for the local wave number and the propagation direction of $b_2$-wave through the outward boundary, and both have variation with the local position change. It is noted that $k_2$ is directly regarded as zero for the overcritical condition.

Due to the induced effect by the body's forward translation, the generated $b_2$-wave system consists of the shorter wave ahead of the body and the longer waves behind the body, as illustrated in Figure 3 for the undercritical condition. At the translating coordinate frame coinciding with the body at speed $U$, when the waves reach point B, the propagation direction rotates from the radial axis by a counterclockwise angle $\theta$.

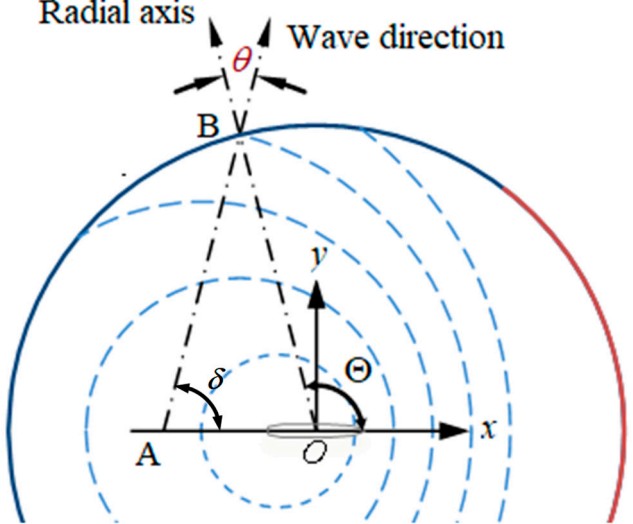

**Figure 3.** Illustration of $b_2$-wave propagation.

The research of Das and Cheung [29] or Yuan, et al. [30] gives Equation (18) to link the local characteristics of the wave number $k_2$, the scattered frequency $\omega_s$ and the propagation angle $\delta(= \Theta - \theta)$ corresponding to the point in the outward boundary.

$$\begin{cases} \frac{U}{V_p} = \frac{\sin\theta}{\sin\Theta} \\ V_p = \frac{\omega_s}{k_2} \\ \omega_s = \omega + Uk_2\cos(\Theta - \theta) = \sqrt{gk_2} \end{cases} \tag{18}$$

Equation (18) is a nonlinear equation set, in which the first equation is derived from the law of sines and the condition of $\overline{AO}/U = \overline{AB}/V_p$, the second one calculates the local phase velocity $V_p$ at B, and the last one is the local dispersion relationship in deep water. An iterative method can provide the local solution of $k_2$ and $\delta$ at the outward boundary point from Equation (18), for satisfying the radiation boundary condition in Equations (16) and (17).

### 2.5. Rayleigh Artificial Damping

As proposed in the proceeding section, the simplified Seto's radiation boundary condition is imposed upstream for all the flow conditions. However, to supplement the insufficiency of imposed upstream boundary conditions, the Rayleigh artificial damping is considered to see its effectiveness, especially for the undercritical conditions ($\tau < 0.25$).

As explained in Nakos [6], introducing Rayleigh viscosity results in an infinitesimal shift of the poles of the linearized free surface boundary condition in complex Fourier plan, and leads to energy dissipation, accordingly. It transforms the encountered oscillatory frequency $\omega$ to $(\omega - i\varepsilon)$, in which $\varepsilon$ is a positive and small value. Furthermore, we can have

$$\omega^2 \rightarrow \omega^2 - 2i\omega\varepsilon + O\left(\varepsilon^2\right) \equiv \omega^2 - i\mu\omega \tag{19}$$

Here, the coefficient $\mu = 2\varepsilon$ is called the Rayleigh artificial damping. The linearized free surface boundary condition in Equation (6) is modified with Rayleigh artificial damping to have expression as follows

$$\begin{aligned} \frac{\partial\varphi}{\partial n}\left(\vec{q}\right) = \Big\{ &\kappa - i\tau(\Phi_{xx} + \Phi_{yy}) - i\mu\kappa \\ &- \Big[2i\tau\Phi_x + \frac{2}{\kappa_0}(\Phi_x\Phi_{xx} + \Phi_y\Phi_{xy}) + \frac{1}{\kappa_0}\Phi_x(\Phi_{xx} + \Phi_{yy}) \\ &- \mu\tau\Phi_x\Big]\frac{\partial}{\partial x} \\ &- \Big[2i\tau\Phi_y + \frac{2}{\kappa_0}(\Phi_x\Phi_{xy} + \Phi_y\Phi_{yy}) + \frac{1}{\kappa_0}\Phi_y(\Phi_{xx} + \Phi_{yy}) \\ &- \mu\tau\Phi_y\Big]\frac{\partial}{\partial y} - \frac{1}{\kappa_0}\left(\Phi_x\frac{\partial}{\partial x} + \Phi_y\frac{\partial}{\partial y}\right)^2 \Big\} \end{aligned} \tag{20}$$

In the present study, Rayleigh artificial damping $\mu$ is assumed to be the spatial varying of the free surface extent for avoiding excessive energy dissipation near the vessel. The formula proposed by Iwashita [32] for $\mu(\tau, R)$ reads as follows:

$$\mu(\tau, R) = \mu_c f_1(\tau) f_2(R) \tag{21}$$

Here, $\mu_c$ presents the critical damping level; and $f_1(\tau)$ and $f_2(R)$ are formulated as the polynomials. The formula of $f_1(\tau)$ is expressed as follows:

$$f_1(\tau) = \begin{cases} 1 & \tau \leq \tau_s \\ 6t^5 - 15t^4 + 10t^3 & \tau_s \leq \tau \leq \tau_e \\ 0 & \tau \geq \tau_e \end{cases} \tag{22}$$

where $t = 1 - (\tau - \tau_s)/(\tau_e - \tau_s)$ and $\tau_s = 0.4$ and $\tau_e = 0.5$. The formula of $f_2(R)$ reads as follows:

$$f_2(R) = \begin{cases} a_4 u^4 + a_3 u^3 + a_2 u^2 & 0 \le u < 1 \\ 1 & u \ge 1 \end{cases} \tag{23}$$

The coefficients in Equation (23) can be determined by applying the constraints of $f_2(0) = f_2'(0) = f_2(1) = f_2'(1) = 0$ and $f_2(0.5) = 0.32$.

## 3. Numerical Implementation

### 3.1. Quadratic B-Spline Scheme

In the present study, all values and derivatives of flow velocity potential are expressed in the biquadratic B-spline scheme, in which the associated basis function $B_j^{(2,2)}$ is designed in terms of coordinates with local reference frame at the $j^{\text{th}}$ quadrilateral panel as the product of two quadratic functions:

$$B_j^{(2,2)}\left(\vec{q}\right) \equiv b_j^{(2)}(\xi) \times b_j^{(2)}(\eta) \tag{24}$$

where $\xi = x - x_j$ and $\eta = y - y_j$, in which $(x_j, y_j)$ is the horizontal coordinate of origin of the local coordinate system at the $j^{\text{th}}$ panel. In Equation (24), the definition of quadratic function $b_j^{(2)}(\xi)$ or $b_j^{(2)}(\eta)$ has the expression as follows,

$$b_j^{(2)}(v) = \begin{cases} \frac{1}{2}\left(\frac{3}{2} + \frac{v}{h_v}\right)^2 & -\frac{3h_v}{2} \le v < -\frac{h_v}{2} \\ \frac{1}{2}\left(\frac{3}{4} - \left(\frac{v}{h_v}\right)^2\right) & -\frac{h_v}{2} \le v < +\frac{h_v}{2} \\ \frac{1}{2}\left(\frac{3}{2} - \frac{v}{h_v}\right)^2 & +\frac{h_v}{2} \le v < +\frac{3h_v}{2} \end{cases} \tag{25}$$

The symbol $v$ (whether for the variable or as the subscript index) in this equation represents either $\xi$ or $\eta$. $h_v$ presents the panel size along the local $\xi$-axis or $\eta$-axis of the $j^{\text{th}}$ panel and provides the nominal length scale. Considering the characteristics of the quadratic basis function, the value and first derivative of the basis function are continuous over the span $3h_v$. Because the supports span over the three panels in the $\xi$ and $\eta$ directions, the variation in characteristics in the $j^{\text{th}}$ panel depends not only on the spline control coefficient of this panel but also on the spline control coefficients of the eight neighboring panels $j_k(k = 1 \sim 4, 6 \sim 9)$, as illustrated in Figure 4.

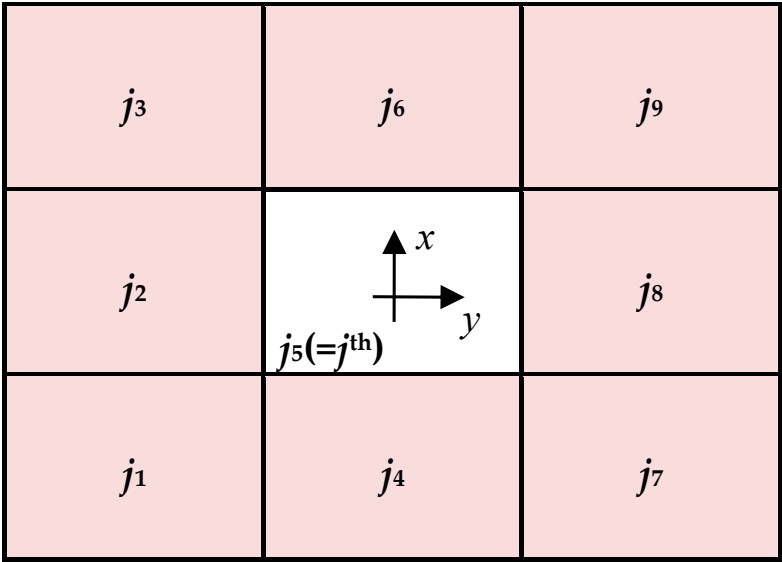

**Figure 4.** Neighboring panels of $j^{\text{th}}$ panel.

Through the present numerical spline scheme, the velocity potential at the point $\vec{q}$ on the $j^{\text{th}}$ panel can be approximated in terms of the highest degree of the two as follows:

$$\varphi\left(\vec{q}\right) = \sum_{k=1}^{9}\left(\sum_{m=0}^{2}\sum_{n=\max(m-2,0)}^{\min(m,2)}\beta_{j_k}^{(n,m-n)}\xi^n\eta^{(m-n)}\right)\sigma_{j_k} \in S_F \cup S_H \tag{26}$$

Here, $\beta_{j_k}^{(n,m-n)}$ represents the spline coefficients of an intermediate $\xi^n\eta^{(m-n)}$ for all neighboring panels with respect to the velocity potential. Their values can be extracted from the associated spatial derivatives: $\frac{\partial\varphi}{\partial x}\left(= \frac{\partial\varphi}{\partial\xi}\frac{\partial\xi}{\partial x} + \frac{\partial\varphi}{\partial\eta}\frac{\partial\eta}{\partial x}\right)$ and $\frac{\partial\varphi}{\partial y}\left(= \frac{\partial\varphi}{\partial\xi}\frac{\partial\xi}{\partial y} + \frac{\partial\varphi}{\partial\eta}\frac{\partial\eta}{\partial y}\right)$ for the further derivation of the normal dipole. As a result, the normal dipole $\frac{\partial\varphi}{\partial n}$ in Equation (6) or (20) can be written in the spline form as follows,

$$\frac{\partial\varphi}{\partial n}\left(\vec{q}\right) = \sum_{k=1}^{9}\left(\sum_{m=0}^{2}\sum_{n=\max(m-2,0)}^{\min(m,2)}\gamma_{j_k}^{(n,m-n)}\xi^n\eta^{(m-n)}\right)\sigma_{j_k} \in S_F \tag{27}$$

where $\gamma_{j_k}^{(n,m-n)}$ stands for the spline coefficients of an intermediate $\xi^n\eta^{(m-n)}$ for the normal dipole expression.

### 3.2. Linear System Construction

The linear complex system for solving unknown splined coefficients is conducted by collecting the resultant discrete boundary integral equations at the collocated points and the employment of end conditions at the extent of truncations and boundaries in the present study.

In practice, the computation extents are to be discretized into the quadrilateral panels for numerical implementation. Herein, those quadrilateral panels over the free surface and the vessel's wetted hull are modeled along the $I$-and $J$-directions, so we can write $N_F = N_F^{(I)} \times N_F^{(J)}$ and $N_H = N_H^{(I)} \times N_H^{(J)}$, in which $N_F$ and $N_H$ account for the number of panels over the free surface and vessel's wetted hull, and the superscript indicates that panel's modeling direction. The numbers of spline coefficients for approximating the flow variation are determined by $M_F = \left(N_F^{(I)} + 2\right) \times \left(N_F^{(J)} + 2\right)$ and $M_H = \left(N_H^{(I)} + 2\right) \times \left(N_H^{(J)} + 2\right)$, based on the present biquadratic spline scheme. Hence, total $M_T(= M_F + M_H)$ spline coefficient unknowns are to be solved in the present problem.

By discretizing the computation extents into $(N_F + N_H)$ quadrilateral panels, we can rewrite the boundary integral equation into a discrete form. Let one field point $\vec{p}$ be collocated in the $l^{\text{th}}$ panel. The discrete boundary integral equation is formulated as follows:

$$2\pi\sum_{k=1}^{9}\mathcal{D}_k^{(l)}\sigma_{l_k}^{(i)} + \sum_{j=1}^{N_F+N_H}\sum_{k=1}^{9}\left(\mathcal{G}_k^{(j)} - \mathcal{H}_k^{(j)}\right)\sigma_{j_k}^{(i)} = \sum_{j=N_F+1}^{N_F+N_H}\mathcal{R}_j \quad i = 1 \sim 6 \tag{28}$$

where the above-mentioned coefficients are defined below.

$$\mathcal{D}_k^{(l)} = \sum_{m=0}^{2}\sum_{n=\max(m-2,0)}^{\min(m,2)}\beta_{l_k}^{(n,m-n)}\xi^n\eta^{(m-n)} \tag{29}$$

$$\mathcal{G}_k^{(j)} = \sum_{m=0}^{2}\sum_{n=\max(m-2,0)}^{\min(m,2)}\beta_{j_k}^{(n,m-n)}Q_j^{(n,m-n)}\left(\vec{p}\right) \tag{30}$$

$$\mathcal{H}_k^{(j)} = \begin{cases} \sum_{m=0}^{2}\sum_{n=\max(m-2,0)}^{\min(m,2)}\gamma_{j_k}^{(n,m-n)}P_j^{(n,m-n)}\left(\vec{p}\right) & j = 1 \sim N_F \\ 0 & j = N_F + 1 \sim N_F + N_H \end{cases} \tag{31}$$

$$\mathcal{R}_j = i\omega\iint_{\Gamma_j}n_i G\left(\vec{p},\vec{q}\right)ds + \iint_{\Gamma_j}m_i G\left(\vec{p},\vec{q}\right)ds \tag{32}$$

In Equations (30) and (31), $P_j^{(m,n)}$ and $Q_j^{(m,n)}$ are the influence coefficients induced by the Rankine source $G$ and the normal dipole $\partial G/\partial n$ over the $j^{\text{th}}$ panel $\Gamma_j$, and defined as follows:

$$P_j^{(m,n)}\left(\vec{p}\right) = \iint\limits_{\Gamma_j} \xi^m \eta^n G\left(\vec{p},\vec{q}\right) ds \tag{33}$$

$$Q_j^{(m,n)}\left(\vec{p}\right) = \iint\limits_{\Gamma_j} \xi^m \eta^n \frac{\partial G}{\partial n}\left(\vec{p},\vec{q}\right) ds \tag{34}$$

In the evaluation of the influence of the $m$-terms in right-hand side of Equation (34), the relation holds as follows

$$\iint\limits_{\Gamma_j} m_i G\left(\vec{p},\vec{q}\right) ds = -\iint\limits_{\Gamma_j} n_i \left(\nabla\Phi\left(\vec{q}\right) \cdot \nabla G\left(\vec{p},\vec{q}\right)\right) ds \tag{35}$$

This is derived from Stokes' theorem with incorporation of the double-body flow as the basis flow in the present study. Details can be found in Nakos [6]. This approach suggests manipulation of the first-order spatial derivatives to replace the second-order spatial derivatives in $m$-terms to raise evaluation accuracy for the $m$-terms. Hence the $\mathcal{R}_j$ is further recast by

$$\mathcal{R}_j = \iint\limits_{\Gamma_j} n_i \left(i\omega G\left(\vec{p},\vec{q}\right) - \nabla\Phi\left(\vec{q}\right) \cdot \nabla G\left(\vec{p},\vec{q}\right)\right) ds \ \ i = 1 \sim 6 \tag{36}$$

The collection of the equations of the discrete boundary integral raised at the panel centroids and corresponding end conditions at the boundaries of extents gives the set of simultaneous equations in matrix form as follows

$$[A]_{M_T \times M_T} [\sigma]_{M_T} = [B]_{M_T} \tag{37}$$

where $[A]$ is a square coefficient matrix; $[\sigma]$ and $[B]$ are the column vectors containing all the unknowns of spline coefficients and force terms of the right-hand side in Equation (28). Once the solution is determined from Equation (37), the flow potential and its normal derivative can be obtained for the subsequent hydrodynamic pressure calculation.

### 3.3. End Conditions

To introduce the appropriate radiation condition for handling the problem of non-unique solution and remedy the underdetermined problem caused by the collocated points on panels, the end conditions are imposed at the truncation of finite extents in free surface and boundary of the vessel's hull.

In the allocation of end conditions, except for the longitudinal truncation at free surface extent, the remaining truncations are imposed with a symmetry condition that makes the flow variation at truncation smooth. As illustrated in Figure 5, the flow variation at truncation point $p = 0.5\, h_{\xi}$, which is the local coordinate relative to the $(l + 1)^{\text{th}}$ panel, is symmetric for both sides and therefore the relation of continuous derivative: $\frac{\partial \varphi}{\partial \xi}(p^+) = \frac{\partial \varphi}{\partial \xi}(p^-) \rightarrow \frac{\partial^2 \varphi}{\partial \xi^2} = 0$, is obtained. At the present spline scheme, we employ the single quadratic spline function, and have $\sigma_l - 2\sigma_{l+1} + \sigma_{l+2} = 0$.

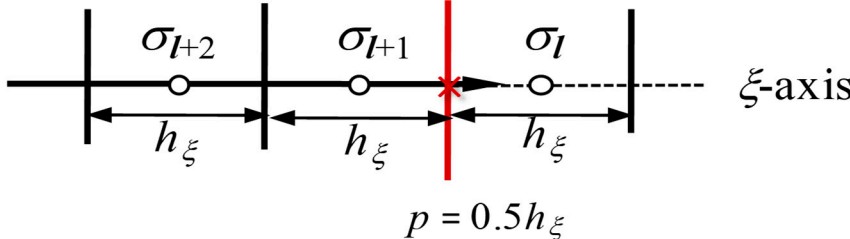

**Figure 5.** Illustration of symmetry boundary condition.

Regarding the end conditions at the longitudinal truncation, only upstream truncation is considered due to its dominance over flow development. Because the technique of the upwind difference or the collocation point shifting is absent from the present B-spline scheme, the radiation conditions proposed in Equations (16) and (17) are employed upstream, and can be expressed in spline form as

$$\sum_{k=1}^{9} \mathcal{L}_{l_k}^{(1)} \sigma_{l_k} = 0 \tag{38}$$

$$\sum_{k=1}^{9} \mathcal{L}_{l_k}^{(2)} \sigma_{l_k} = 0 \tag{39}$$

where the associated coefficients $\mathcal{L}_{l_k}^{(1)}$ and $\mathcal{L}_{l_k}^{(2)}$ are defined by

$$\mathcal{L}_{l_k}^{(1)} \equiv \sum_{m=0}^{2} \sum_{n=\max(m-2,0)}^{\min(m,2)} \left[ \beta_{l_k}^{(n,m-n)} \left( \frac{\partial}{\partial x} + ik_2 \cos \delta \right) \left( \frac{\partial}{\partial x} - i\kappa_0 \tau \right) \xi^n \eta^{(m-n)} \right] \tag{40}$$

$$\mathcal{L}_{l_k}^{(2)} \equiv \sum_{m=0}^{2} \sum_{n=\max(m-2,0)}^{\min(m,2)} \left[ \beta_{l_k}^{(n,m-n)} \left( \frac{\partial}{\partial x} + ik_2 \cos \delta \right) \left( \frac{\partial}{\partial x} - i\kappa_0 \tau \right)^2 \xi^n \eta^{(m-n)} \right] \tag{41}$$

### 3.4. Solving Characteristics of Local Scattered Waves

An efficient way to obtain local wave characteristics, including the wave number $k_2$ and the local propagating direction $\delta$ required in Equations (16) and (17), is proposed to replace the traditional iteration way of solving Equation (18). Stationary phase method on Bessho three-dimensional translating-pulsating Green function, which is a single integral form satisfying the linearized free-surface boundary condition and radiation condition at infinity, is considered to have the set of parametric equations for the constant-phase curves as Equation (42). The detailed formulation of the Bessho Green function can refer to the works in [41,42].

$$\begin{cases} \kappa_0 x = \frac{k_2 \sin \vartheta + \dot{k}_2 \cos \vartheta}{k_2^2} \psi(k_2, \vartheta) \\ \kappa_0 y = \frac{k_2 \sin \vartheta - \dot{k}_2 \cos \vartheta}{k_2^2} \psi(k_2, \vartheta) \end{cases} \tag{42}$$

where $\psi(k_2, \vartheta)$ is the phase function of the wave number $k_2$ and variable $\vartheta$. Both $k_2$ and its derivative $\dot{k}_2$ are the functions of $\vartheta$. Here $\vartheta$ is the integral variable used in the Bessho Green function for the integral of $-\pi \le \vartheta \le 0$ for the undercritical condition ($\tau < 0.25$).

$$k_2 = \frac{\kappa_0}{2 \cos^2 \vartheta} \left( 1 + 2\tau \cos \vartheta - \sqrt{1 + 4\tau \cos \vartheta} \right) \tag{43}$$

Let (x, y) represents the point B at outward boundary in Figure 3, which is the generated wave pattern by a translating body at undercritical condition. As indicated in the top diagram of Figure 6, it presents the generated wave patterns, including the circular waves ($b_2$-waves) and two sets of divergent and transverse waves ($a_1$- and $a_2$-waves), for

the undercritical condition ($\tau < 0.25$). When the circular wave approaches the point B, at which the local wave direction deflects from the radial axis with the angle of $\theta$, due to the Doppler-shifting effect. We can correct the circular wave propagating direction to angle of $\delta_B (= \Theta_B - \theta_B)$. $\Theta_B$ corresponds to the polar angle at B, and it can be obtained by the resultant relationship of $(\kappa_0 y / \kappa_0 x)$:

$$\tan \Theta = \frac{k_2 \sin \vartheta - \dot{k_2} \cos \vartheta}{\dot{k_2} \sin \vartheta + k_2 \cos \vartheta} \tag{44}$$

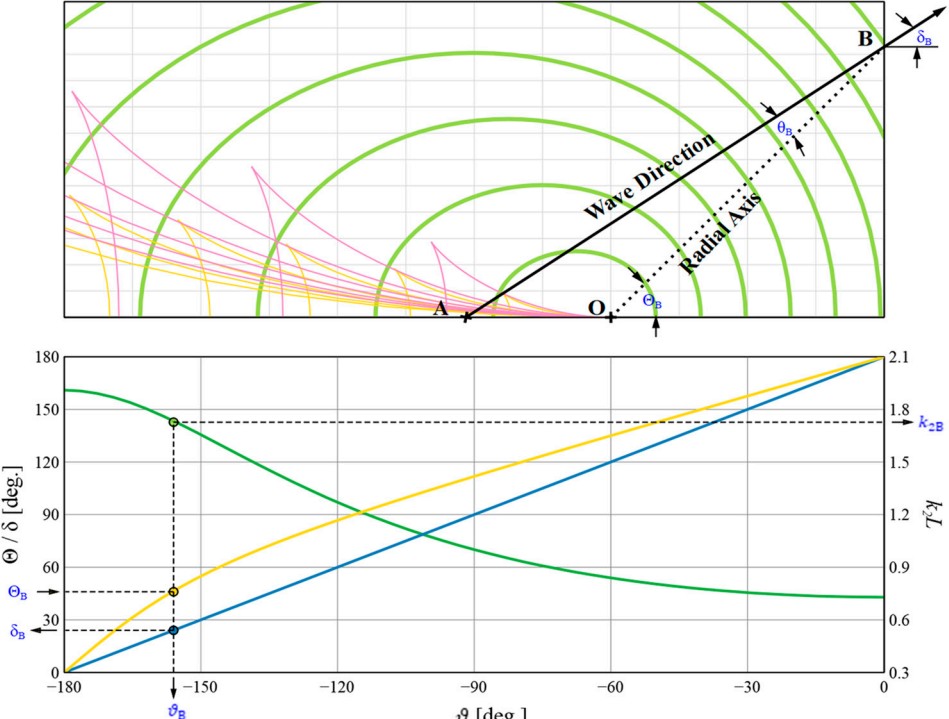

**Figure 6.** Relationship among the local scattered wave characteristics. (**Top**) Green line: $b_2$-waves (circular waves), Pink line: $a_1$-waves (divergent & transverse waves), Orange line: $a_2$-waves (divergent & transverse waves); (**Bottom**) Green curve: $k_2$ vs. $\vartheta$, Blue curve: $\delta$ vs. $\vartheta$, Orange curve: $\Theta$ vs. $\vartheta$.

The propagating direction $\delta$ can be obtained from Equation (18) by

$$\cos \delta = \frac{\sqrt{gk_2} - \omega}{Uk_2} \tag{45}$$

The consequence from Equations (44) and (45) can be expressed in function of $\vartheta$ to construct the relationship between the wave number $k_2$ and propagating direction $\delta$ at local position $\Theta$, as presented in bottom diagram of Figure 6. Substituting $\vartheta$ varying from $-\pi$ to 0 into the Equations (44) and (45) could have a relationship as the diagram in Figure 6. Once specific $\vartheta$, which corresponds to $\Theta_B$, is pointed out, the associated solution of $k_2$ and $\delta$ succeeds. Compared to the iteration method, the proposed alternative is explicit, and apparently efficient to determine the radiation boundary condition at various upstream points.

### 3.5. Induced Flow by the Submerged Source

The flow field induced by an oscillatory source translating under water has been evaluated by the present Rankine source method with the proposed radiation technique to investigate its feasibility under overcritical and undercritical conditions. The associated computation configuration is presented in Figure 7, in which the source is moving at the

speed $U$ and the depth of $d$. The investigation arises by comparing the present evaluation against the analytic solution obtained using the Bessho form three-dimensional translating-pulsating-source Green function, in which the steepest descent numerical integration proposed by [42] is utilized to obtain the analytic solution in this paper.

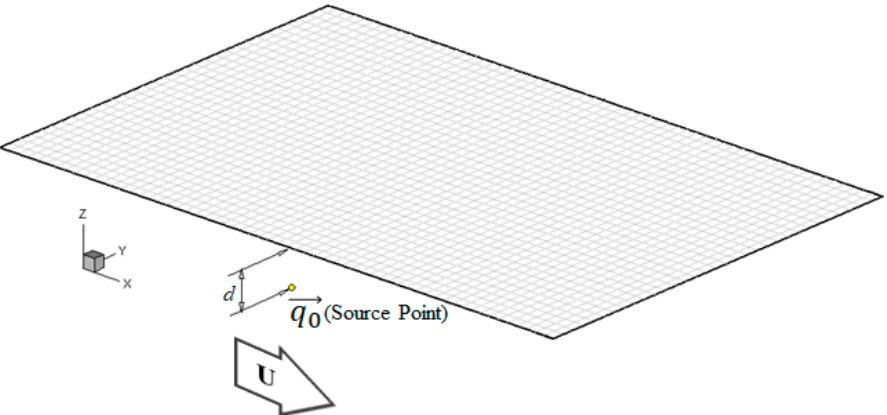

**Figure 7.** Computation configuration for the moving source point under water.

Let field point $\vec{p}$ be allocated in the $l^{\text{th}}$ quadrilateral panel. The associated discrete boundary integral followed by Equation (28) is expressed as

$$2\pi \sum_{k=1}^{9} \mathcal{D}_k^{(l)} \sigma_{l_k} + \sum_{j=1}^{N} \sum_{k=1}^{9} \left( \mathcal{G}_k^{(j)} - \mathcal{H}_k^{(j)} \right) \sigma_{j_k} = G\left( \vec{p}, \vec{q_0} \right) \tag{46}$$

in which the right-hand side presents the velocity potential induced by the Rankine source $\vec{q_0}$ at $\vec{p}$. Once the spline coefficients in Equation (46) are solved, the unsteady velocity potential $\varphi$ over the free surface can be evaluated. The induced unsteady flow field by an oscillating and translating singularity at $(0, 0, -L/10)$ with Froude number of $F_r = 0.2$ is simulated by the present method. For convenience, only real part of the unsteady velocity potential is considered in the following discussion.

The associated comparison for the evaluated flow field by present Rankine source scheme without Rayleigh damping ($\mu_c = 0.0$) against the analytical solution at the undercritical condition of $\tau = 0.2$, attributing from the oscillatory frequency at $\omega = 1.0$, is shown in Figure 8, and the counter solution, introducing the Rayleigh damping ($\mu_c = 0.6$), is presented in Figure 9. The improvement in comparing Figures 8 and 9 can be observed at the pattern of circular waves ahead of the disturbance, and it reveals that the Rayleigh damping supplies the energy dissipation that is insufficient by Seto's radiation boundary condition only. Furthermore, the employed technique of radiation condition with Rayleigh artificial damping yields a complete and accurate flow around the disturbed singularity at this undercritical condition, although the crests are less or more damped.

Figure 10 presents the flow patterns by the Rankine source scheme and the analytical solution at $\tau = 0.347$, which attributes from oscillatory frequency of $\sqrt{3}$ and Froude number of $F_r = 0.2$. The associated comparison on the generated divergent wave and transverse wave is at high agreement. Therefore, the good results in Figure 9 ($\tau = 0.2$) and Figure 10 ($\tau = 0.347$) conclude that the proposed Rankine source scheme, based on Rayleigh artificial damping in additional to simplified Seto's radiation boundary condition, is acceptable for the undercritical and overcritical conditions and can be further applied to the real ship analysis.

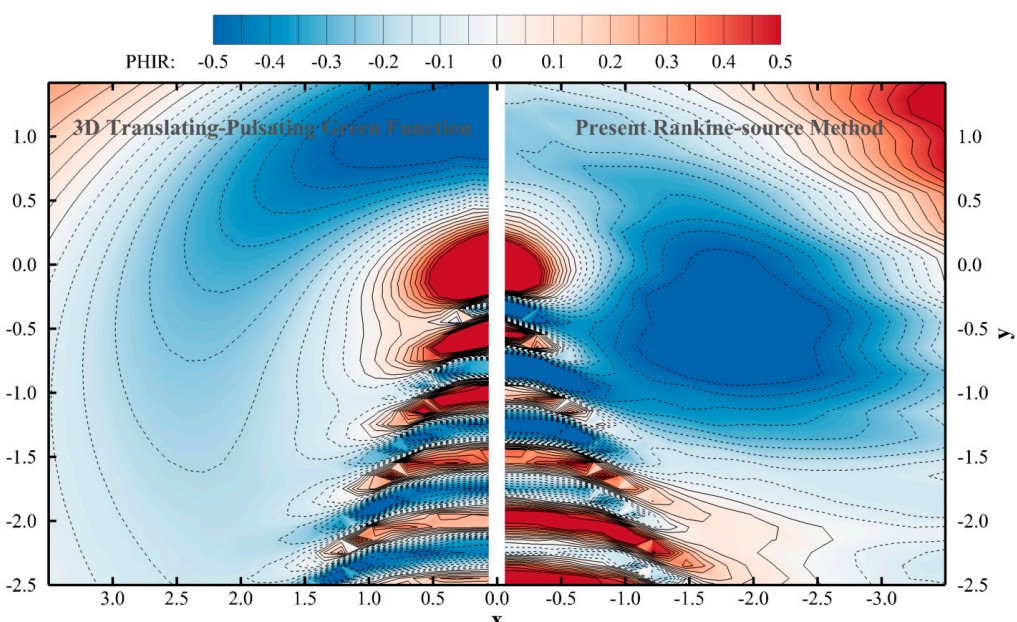

**Figure 8.** $\mathbb{R}[\varphi]$ by Green function and Rankine source methods at $\kappa L = 1.0$, $F_r = 0.2$, $\tau = 0.2$ and without Rayleigh damping $\mu_c = 0.0$ [39].

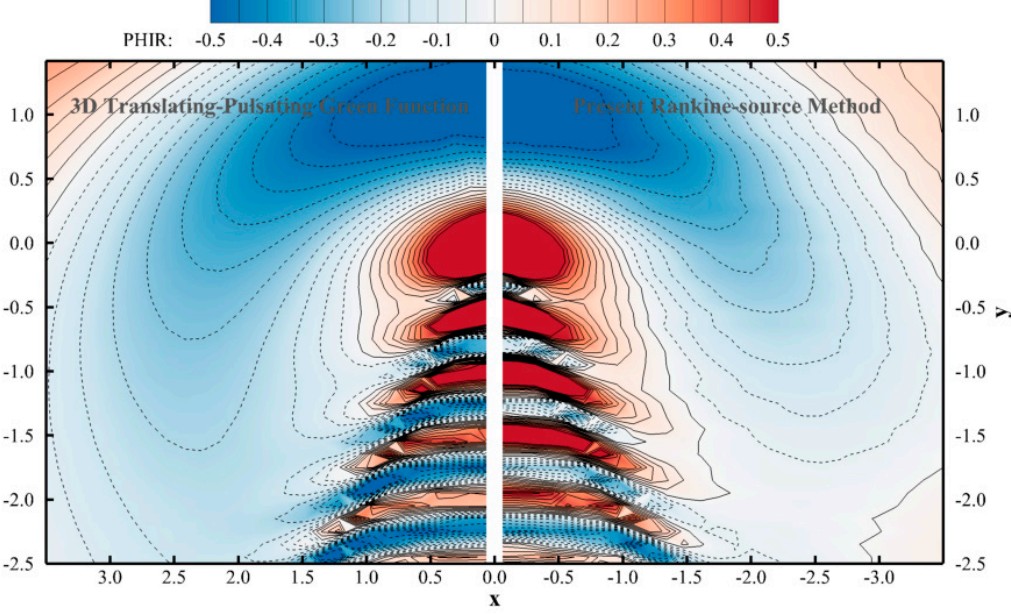

**Figure 9.** $\mathbb{R}[\varphi]$ by Green function and Rankine source methods at $\kappa L = 1.0$, $F_r = 0.2$, $\tau = 0.2$ and with Rayleigh damping $\mu_c = 0.6$ [39].

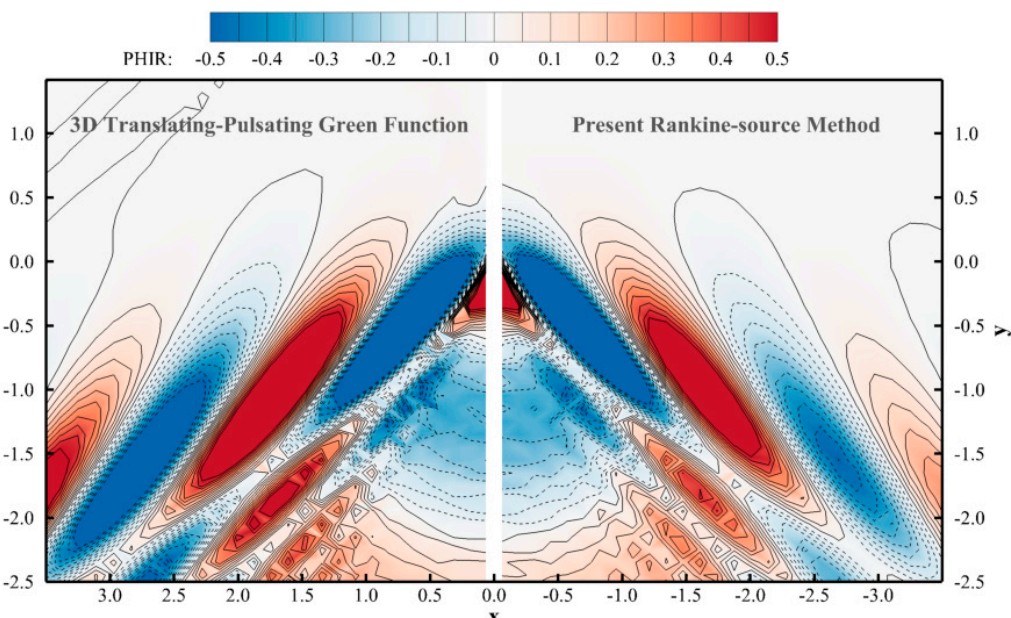

**Figure 10.** $\mathbb{R}[\varphi]$ by Green function and Rankine source methods at $\kappa L = 3.0$, $F_r = 0.2$, $\tau = 0.347$ and with Rayleigh damping $\mu_c = 0.6$ [39].

Due to the attempt to have an appropriate $R_c$ for Rayleigh artificial damping distribution, we investigated the difference between the solutions by the present Rankine source scheme and analytical method. The mean difference between both solutions is denoted by $e$, and has the definition in Equation (47), in which $\phi$ and $\varphi$ denote the corresponding evaluated and analytical flow potential at arbitrary point $\vec{p_i}$ in free surface and $M$ points are considered in total. The superscript * stands for the conjugate of complex number. Because the surrounding flow field of disturbance singularity $\vec{q_0}$ is our major concern, the specified region taken into account of the flow solution difference investigation is at a rectangular extent, bounded by $L$ to $-2L$ in longitudinal and from 0 to $L$ in transversal direction, with respect to $\vec{q_0}$.

$$e = \frac{100\%}{M} \sum_{i=1}^{M} \sqrt{\frac{(\varphi_i - \phi_i)(\varphi_i - \phi_i)^*}{\phi_i \phi_i^*}} \tag{47}$$

The induced flow fields at the undercritical condition of $\tau = 0.2$ are solved with various parametric sets of critical Rayleigh damping ($\mu_c$) and critical radial distance ($R_c$), against the corresponding analytical solution. The resultants of mean difference at various ($\mu_c$, $R_c$) are summarized in Figure 11, in which the employed $\mu_c$ covers from 0.0 to 0.9, and $R_c/L$ is given at 1.5, 2.0 and 2.5. As presented in Figure 11, the value of mean difference obviously decreases with the increasing Rayleigh damping and trends around the level of $e = 30\% \sim 35\%$, even at different $R_c/L = 1.5$, 2.0 or 2.5. The optimal parametric set in this investigation is found to have $\mu_c = 0.6$ and $R_c/L = 2.0$, and the comparisons of evaluated velocity potentials against the analytic one along several longitudinal lines at $y/L = 0.06$, 0.18, 0.30 and 0.42 are considered. As shown in Figure 12, good agreement between the evaluated and analytic solutions in both real and image parts is noted, especially at the area ahead of the disturbance. Therefore, this parametric set of ($\mu_c$, $R_c$) can then be further employed for solving a radiation flow problem resulting from a real ship.

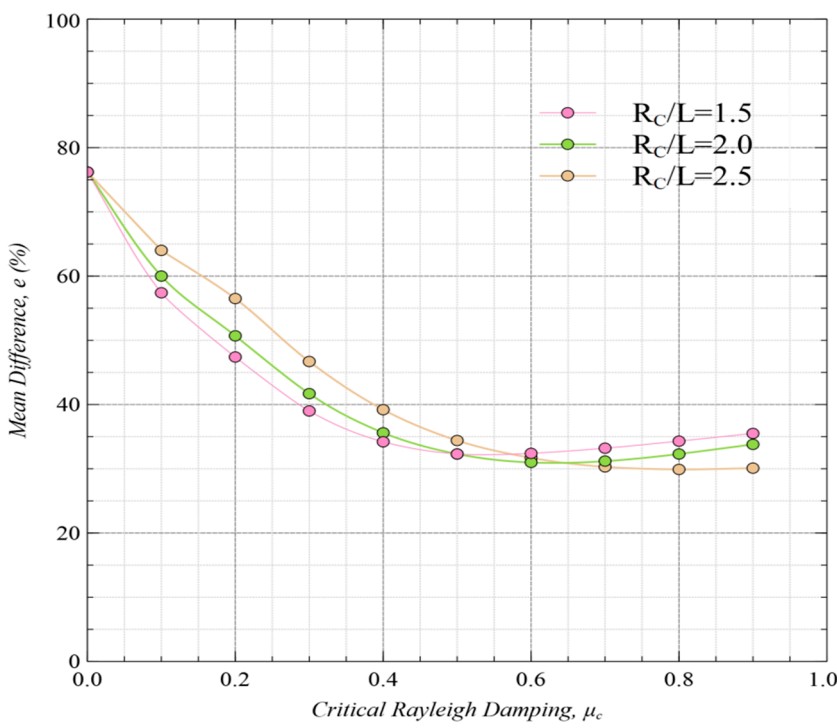

**Figure 11.** $\mathbb{R}[\varphi]$ mean difference at various $(\mu_c, R_c)$.

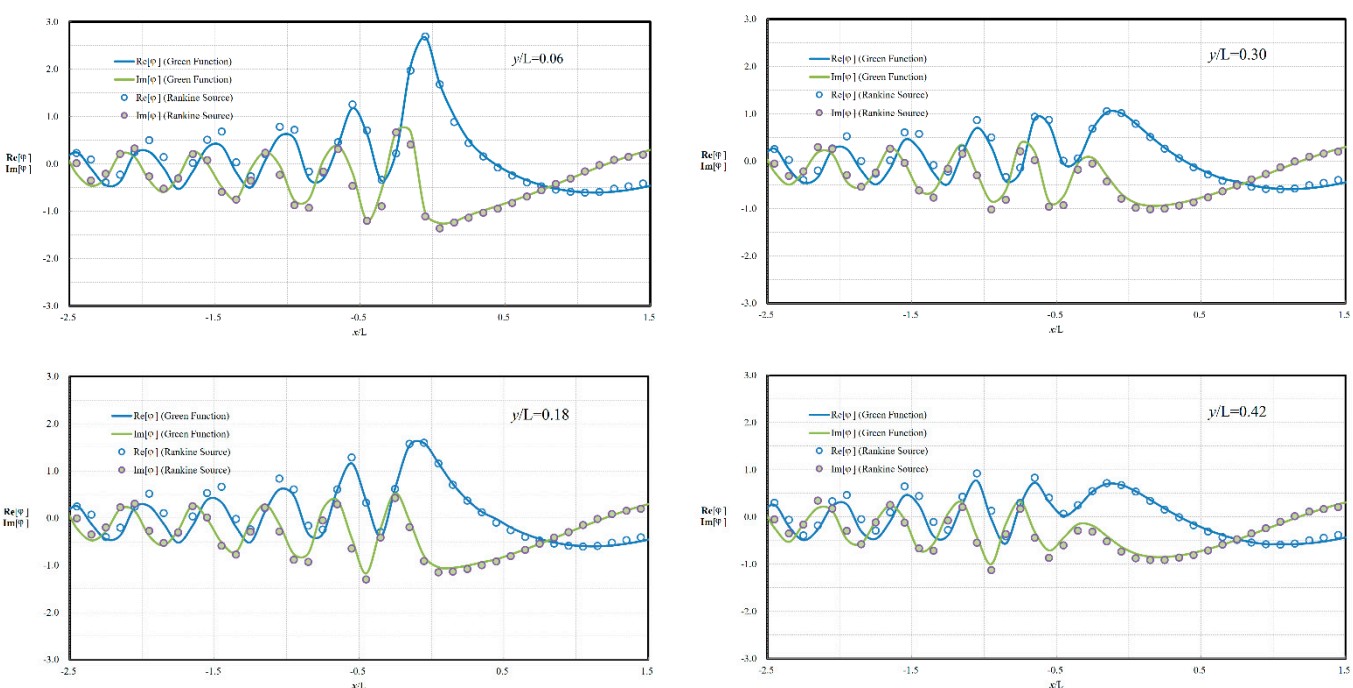

**Figure 12.** Comparison of induced velocity potential along lines.

## 4. Results and Discussion

The attempt to investigate the applicability of present biquadratic B-spline Rankine source scheme on both undercritical and overcritical conditions, the RIOS (Research Initiative on Ocean Ships) bulker is selected as the numerical model for calculations and the results are compared with the experiment data and numerical prediction by Iwashita, et al. [32]. The ship model seakeeping tests of RIOS bulker were conducted in the Research Institute for Applied Mechanics, Kyushu University and its principal dimensions of the model are shown in Table 1.

**Table 1.** Principal dimensions of model.

| Length | $L$ | 2.400 | (m) |
|---|---|---|---|
| Breath | $B$ | 0.400 | (m) |
| Draft | $D$ | 0.128 | (m) |
| Displacement | $\triangledown$ | 0.0983 | (m$^3$) |
| Center of gravity (*x*-axis) | $x_G$ | 0.051 | (m) |
| Center of gravity (*z*-axis) | $y_G$ | −0.020 | (m) |
| Gyration radius in pitch | $K_{yy}/L$ | 0.250 | |

In addition to the capacity of the facility (such as carrier, load cell and so on), the available experiment data range in a seakeeping test is mainly restricted by the size of the test basin, because the near wave field around the ship model might be interfered with by the wave reflection from the side wall of the basin, especially for the longer wave lengths. This wave interference causes inaccuracy in measurement, and it is difficult to avoid for cases at $\tau \leq 0.25$. Therefore, the existing associated numerical results are taken into account in validation of the author's proposed method.

Figure 13 shows the modeled panels representing the hull surface under still water, and 1258 quadrilateral panels in half hull surface are employed in the computation. Under the satisfaction of stability requirement, the total 6498 quadrilateral panels allocated on the free surface computation domain are adopted, which is a rectangular extent bounded by $-2.5L \leq x \leq 1.5L$ in *x*-direction and $0 \leq y \leq 4.5L$ in positive *y*-direction, as shown in Figure 14. The truncation distance in *y*-direction is chosen to allow for sufficient space for developing the wave system such that the edges of the sector do not intersect the transverse outward boundary.

The resulting comparison for the coupled hydrodynamic coefficients induced by the oscillating model in heave and pitch modes translating at $F_r = 0.18$ are presented in Figures 15–18, where two axes of abscissa using $\kappa L$ (dimensionless wave number) and $\tau$ are adopted for convenient reading. In calculating the results, the oscillatory wave numbers of $\kappa L$ ranging from 0.4 to 40.0 are considered to cover the flow conditions corresponding to $\tau < 0.25$ and $\tau > 0.25$. The critical Rayleigh damping $\mu_c = 0.6$ and critical distance $R_c = 2.0$ are adopted in the calculations at various flow conditions.

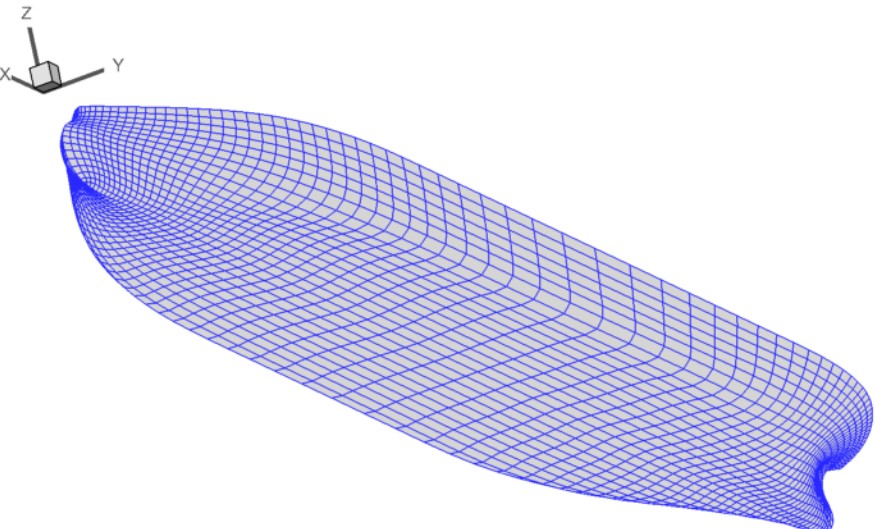

**Figure 13.** Panels modeled on the model hull [32].

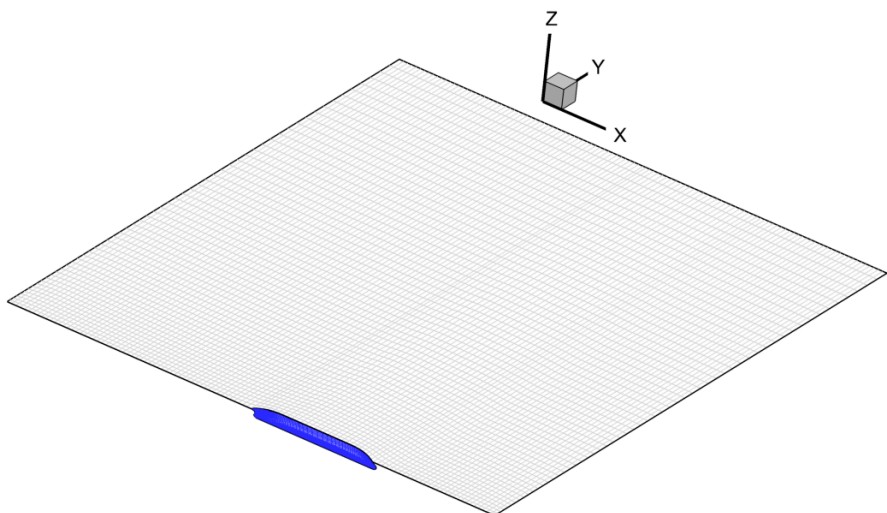

**Figure 14.** Panels modeled over the free surface.

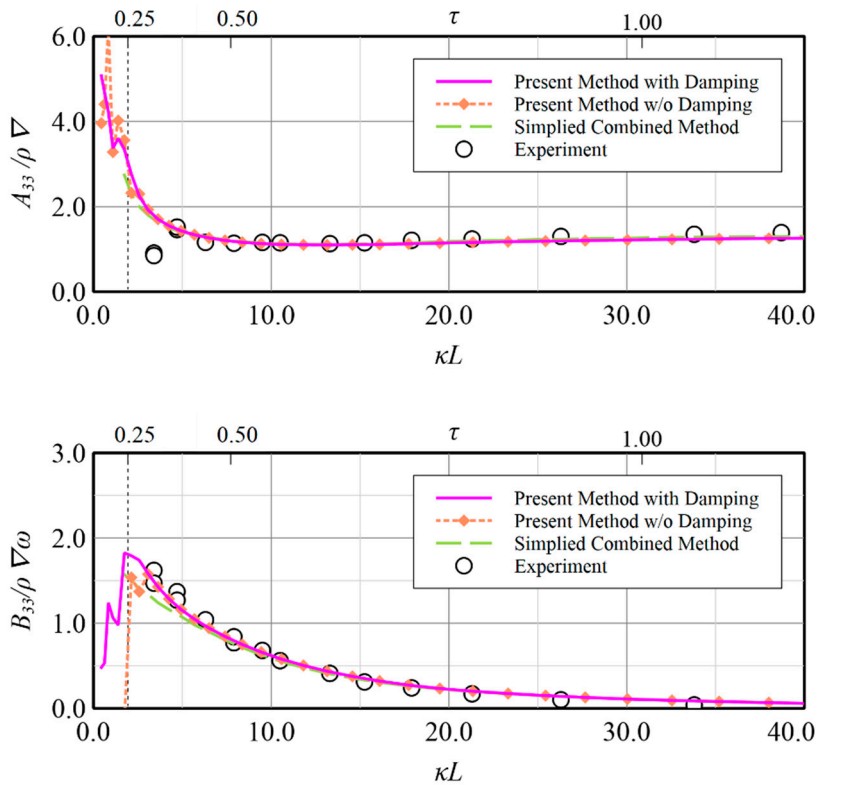

**Figure 15.** Heave-induced-heave added mass and damping coefficient at $F_r = 0.18$.

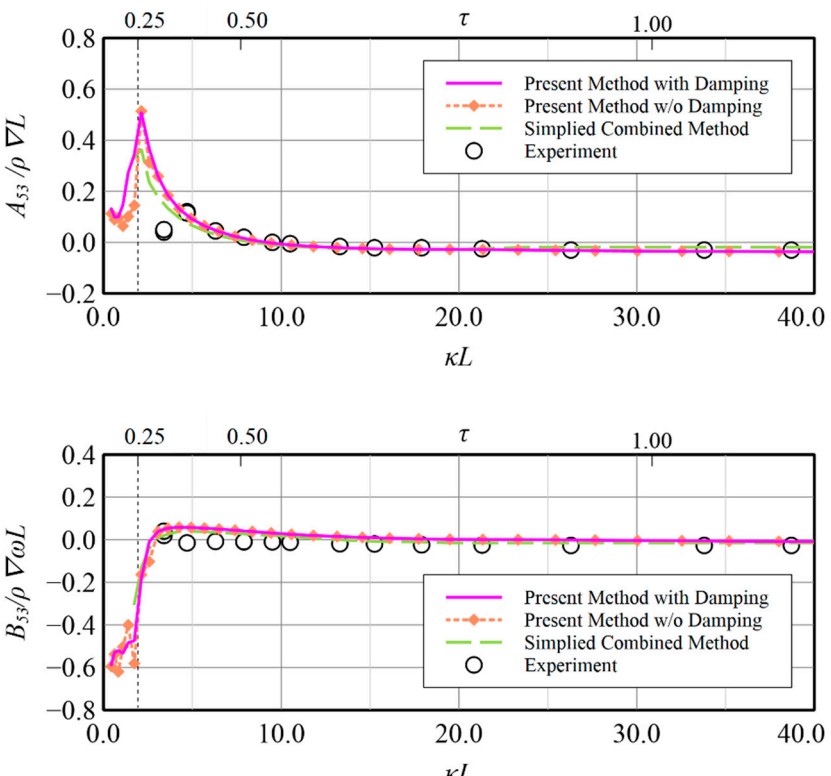

**Figure 16.** Heave-induced-pitch added mass and damping coefficient at $F_r$ = 0.18.

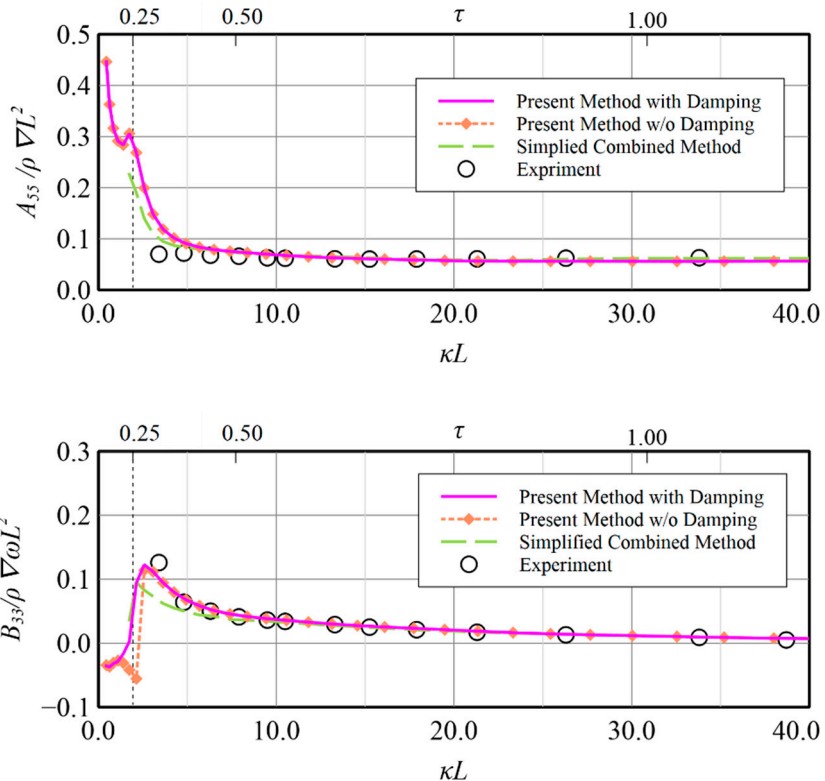

**Figure 17.** Pitch-induced-pitch added mass and damping coefficient at $F_r$ = 0.18.

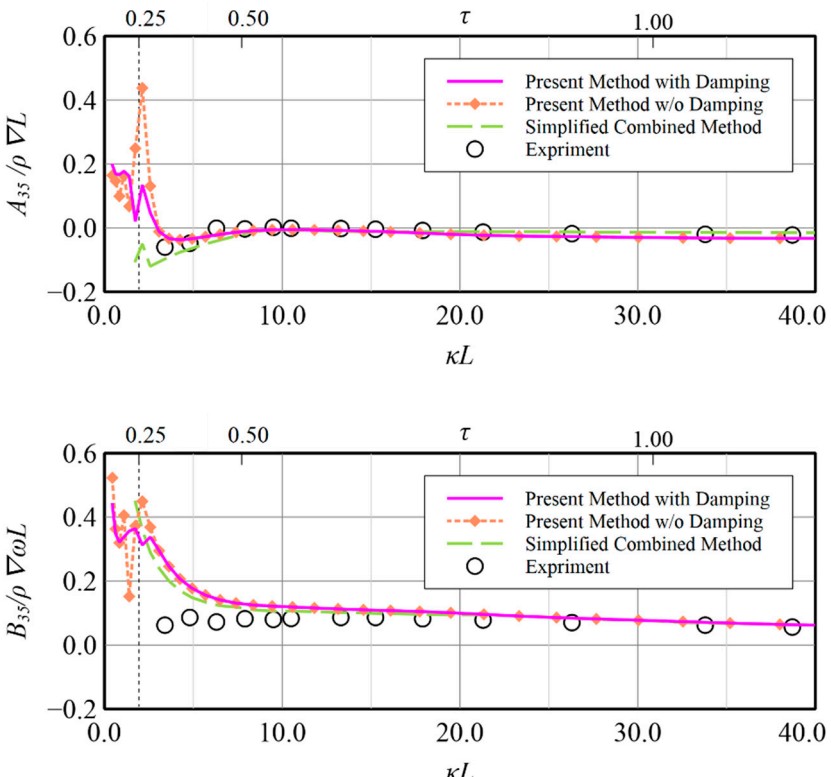

**Figure 18.** Pitch-induced-heave added mass and damping coefficient at $F_r = 0.18$.

It is worth noting that the explicit radiation boundary condition imposed upstream is an essential implementation for producing wave radiation in the present scheme structure. The calculations with and without damping are shown to investigate the contributions due to Rayleigh artificial damping in additional to inherent radiation boundary conditions.

In Figures 15 and 16, due to the forced heave oscillating motion, the heave-induced-heave and heave-induced-pitch added mass and damping coefficients are presented, respectively. Generally good agreement of the present method in comparison with either the experiments or the other numerical solutions based on the simplified combined method [32] are found. Only some discrepancies occurring in heave-induced-heave added mass, i.e., $A_{33}$ in Figure 15, at low wave number around $\kappa L = 3.0$ and in heave-induced-pitch damping, i.e., $B_{53}$ in Figure 16, at $\kappa L < 10.0$, which indicates that numerical solutions are overestimated against the experiment. Furthermore, the high similarity of trend and quantity reveals that the present method is capable of providing a reasonable solution as the simplified combined method does. The apparent difference between the solutions with and without Rayleigh damping is only found at the region of low wave number, i.e., $\tau < 0.25$, which indicates Rayleigh artificial damping indeed makes the results more stable than those without imposing Rayleigh artificial damping which generally show large fluctuation phenomena.

The numerical solutions and experimental data for the coupled pitch-induced-pitch and pitch-induced-heave added mass and damping coefficient by the forced pitch oscillating motion are presented in Figures 17 and 18, respectively. Excellent agreement is generally seen in comparison with the experiment, although some overestimated pitch-induced-heave, i.e., $B_{35}$, at $\kappa L < 10.0$ is noted. Because no experiment is available, validation of the present method at low $\tau$ is specifically compared with the existing results from the simplified combined method. As shown in Figure 17, the differences among the pitch-induced-pitch results by numerical methods are small. Furthermore, it is noted that the drop-down phenomenon near $\tau = 0.25$ in pitch-induced-pitch damping coefficient, i.e., $B_{55}$, by the present method without damping is improved as smooth one. The effectiveness of the damping is that the coupled pitch-induced-heave results (Figure 18) is also found,

since the oscillation at low $\tau$ is depressed in the result with damping. At $\tau = 0.25$, a peak in the pitch-induced-heave added mass, i.e., $A_{35}$, result by the simplified combined method is noted. It is also seen in the results by the present method, although its peak value is higher. Generally, the same conclusions can be drawn for the hydrodynamic coefficients for the forced pitch oscillating motion mode, like the previous observation in forced heave motion mode.

From the above comparisons either with experiment and other numerical method [32] in vertical motion modes, the validity of the present method by constructing the resultant boundary integral equation by employing simplified Seto's explicit conditions and Rayleigh artificial damping to accommodate wave radiation has been confirmed and application on the rest motion modes i.e., surge, sway, roll and yaw, may also be regarded to be workable although no related data can be compared. Some benefits on the calculation results by applying the present technique for these modes are shown in Figures 19–26 for reference. The comparisons in Figures 19–22, in which the induced diagonal added mass and damping coefficient at motion modes of surge, sway, roll and yaw are presented, illustrate that introducing Rayleigh artificial damping suppresses the fluctuation phenomena of results without damping in low frequency regions and makes the solutions more stable, i.e., $\tau < 0.25$. A similar conclusion can also be drawn for the coupled horizontal motion modes, i.e., roll-induced-sway, roll-induced-yaw, sway-induced-roll and yaw-induced-roll, as presented in Figures 23–26, respectively.

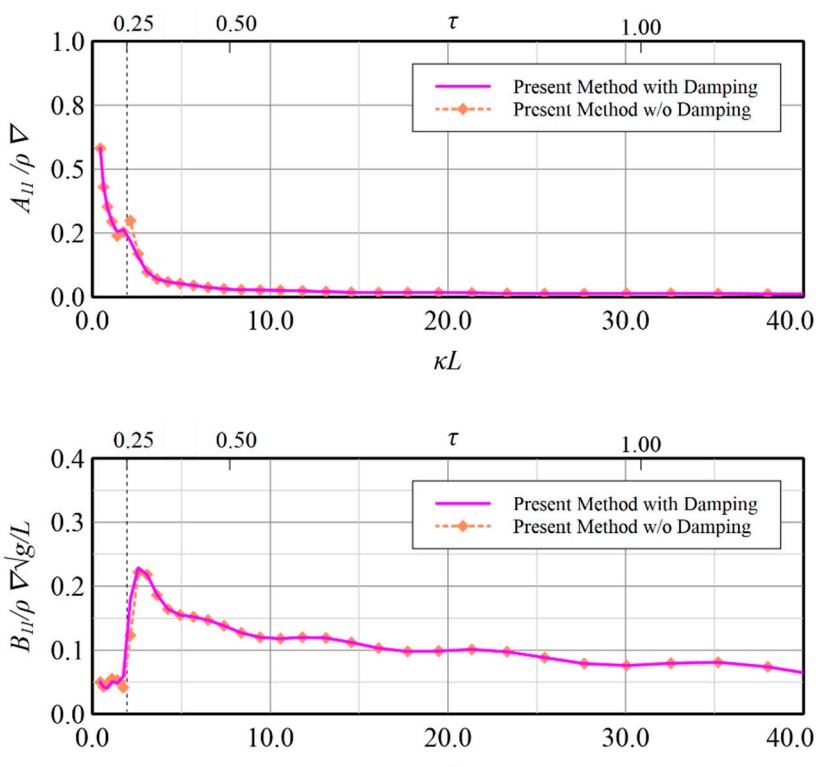

**Figure 19.** Surge-induced-surge added mass and damping coefficient at $F_r = 0.18$.

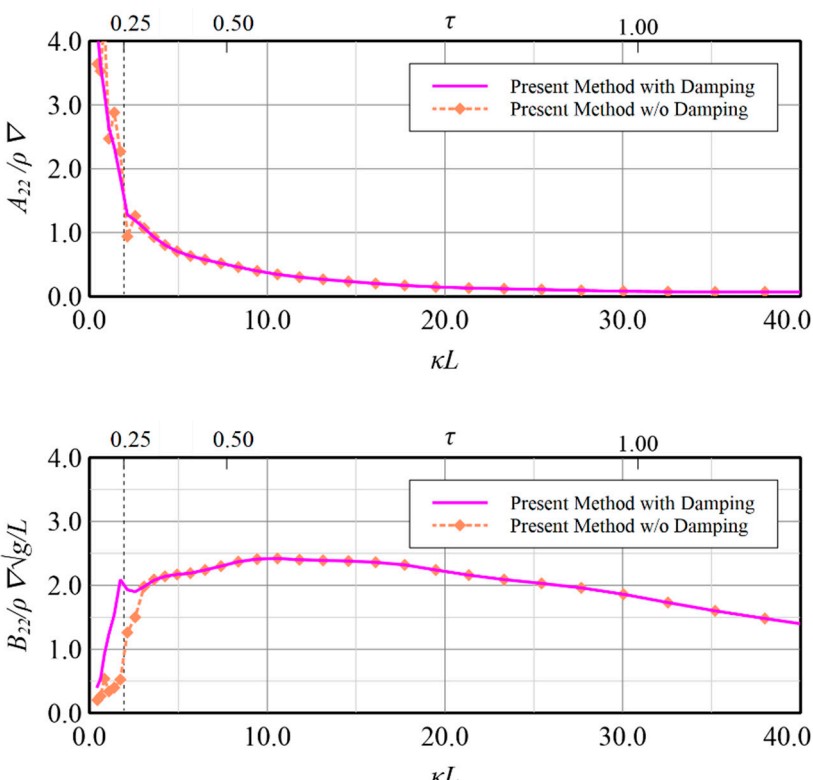

**Figure 20.** Sway-induced-sway added mass and damping coefficient at $F_r = 0.18$.

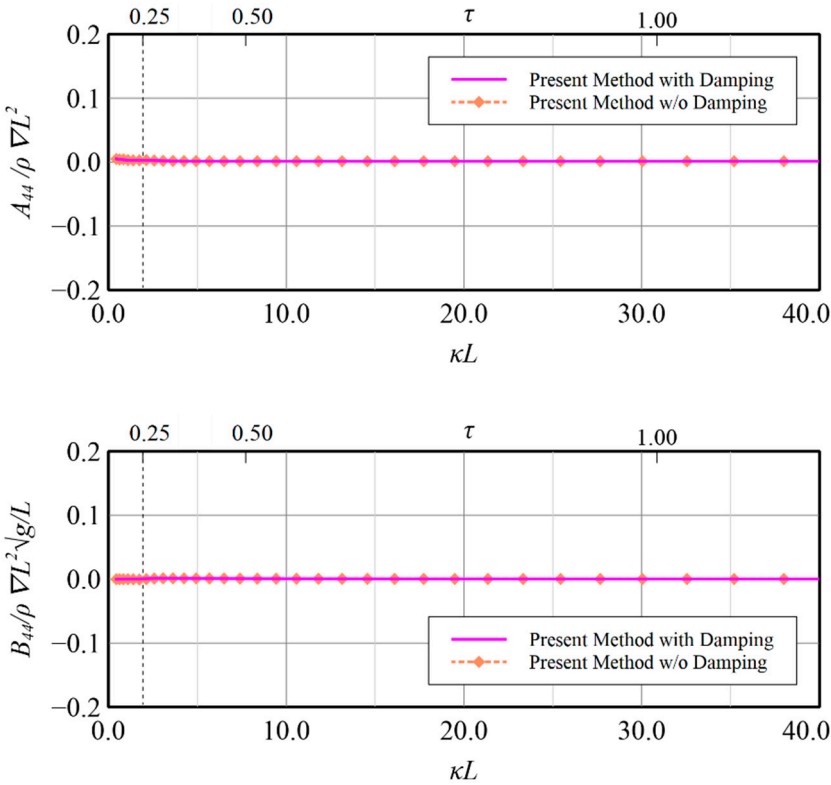

**Figure 21.** Roll-induced-roll added mass and damping coefficient at $F_r = 0.18$.

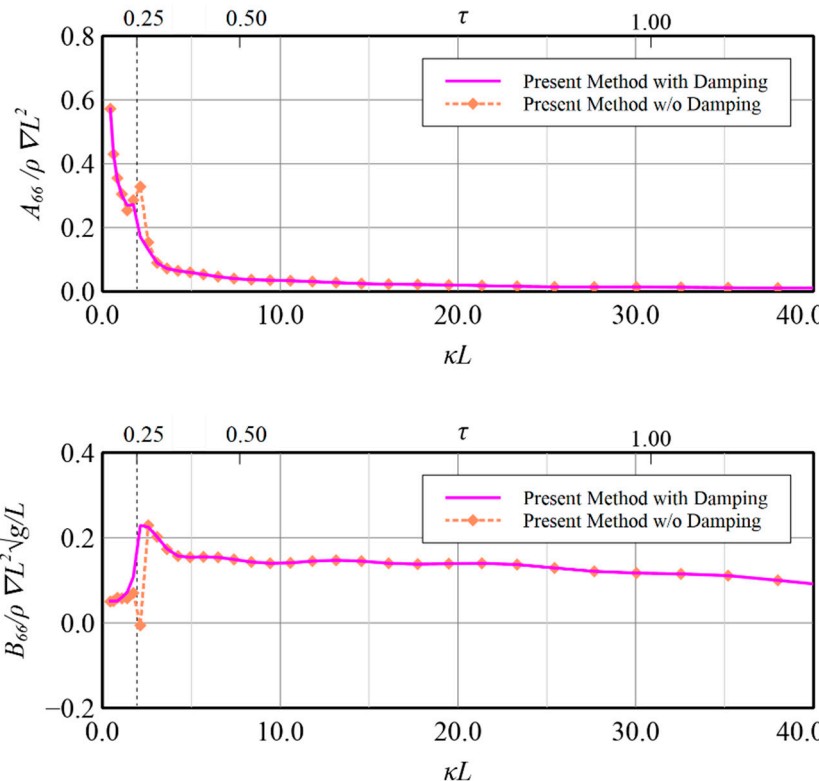

**Figure 22.** Yaw-induced-yaw added mass and damping coefficient at $F_r = 0.18$.

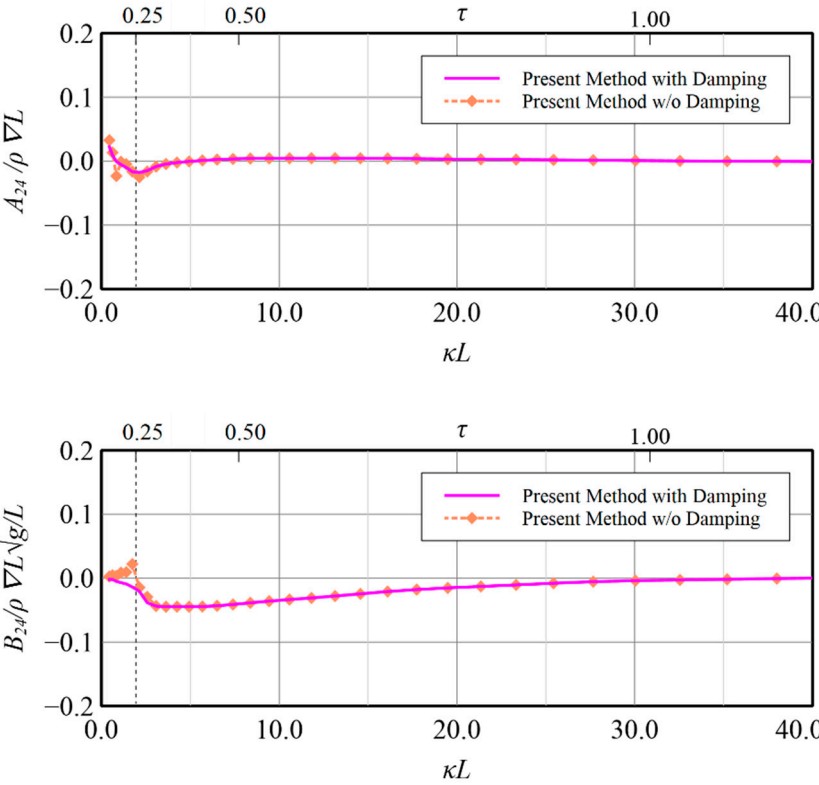

**Figure 23.** Roll-induced-sway added mass and damping coefficient at $F_r = 0.18$.

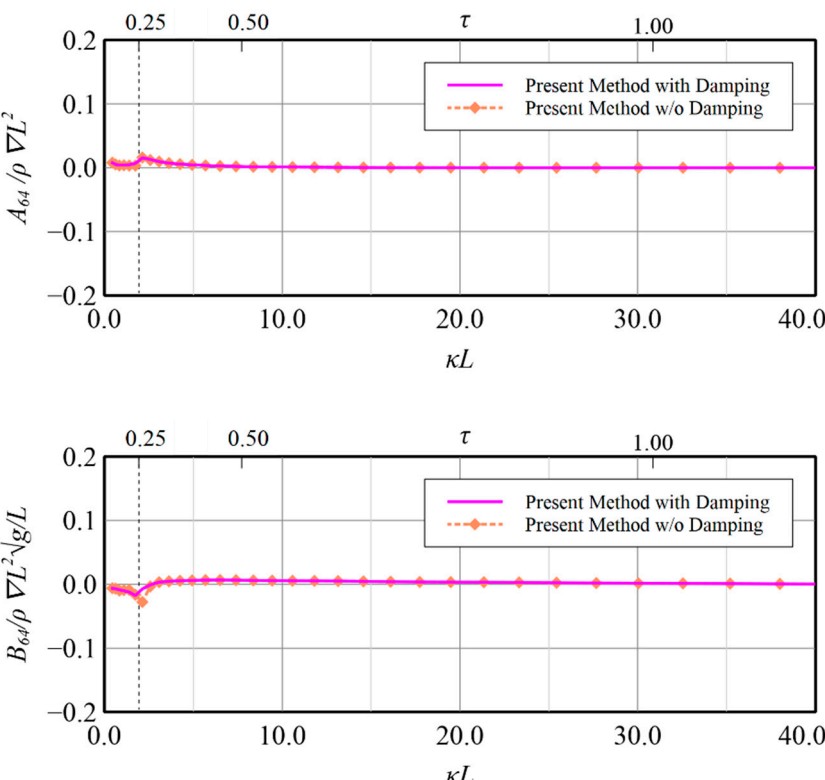

**Figure 24.** Roll-induced-yaw added mass and damping coefficient at $F_r = 0.18$.

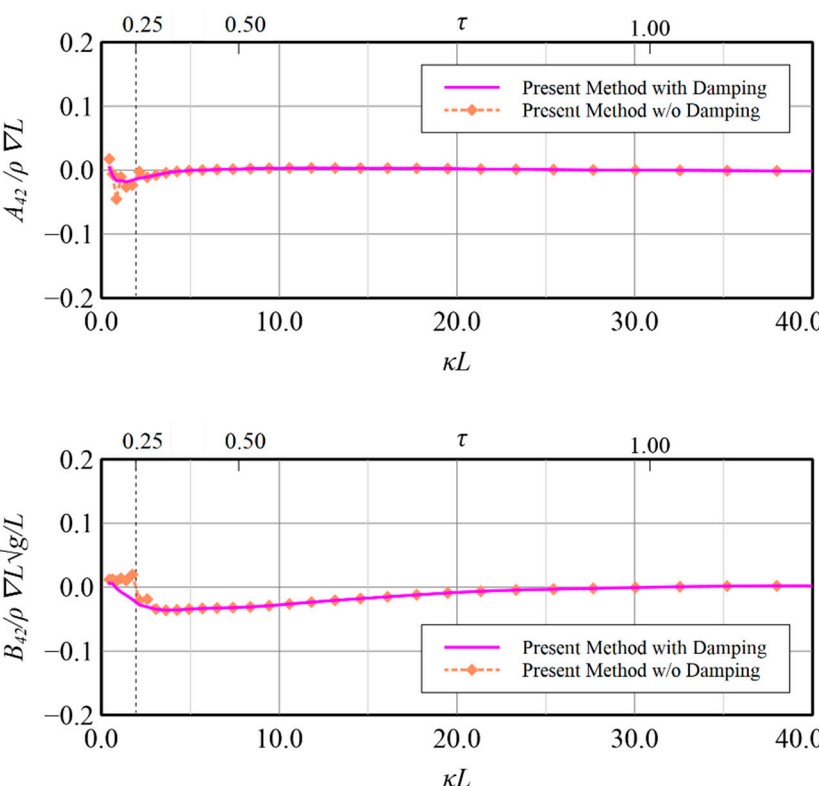

**Figure 25.** Sway-induced-roll added mass and damping coefficient at $F_r = 0.18$.

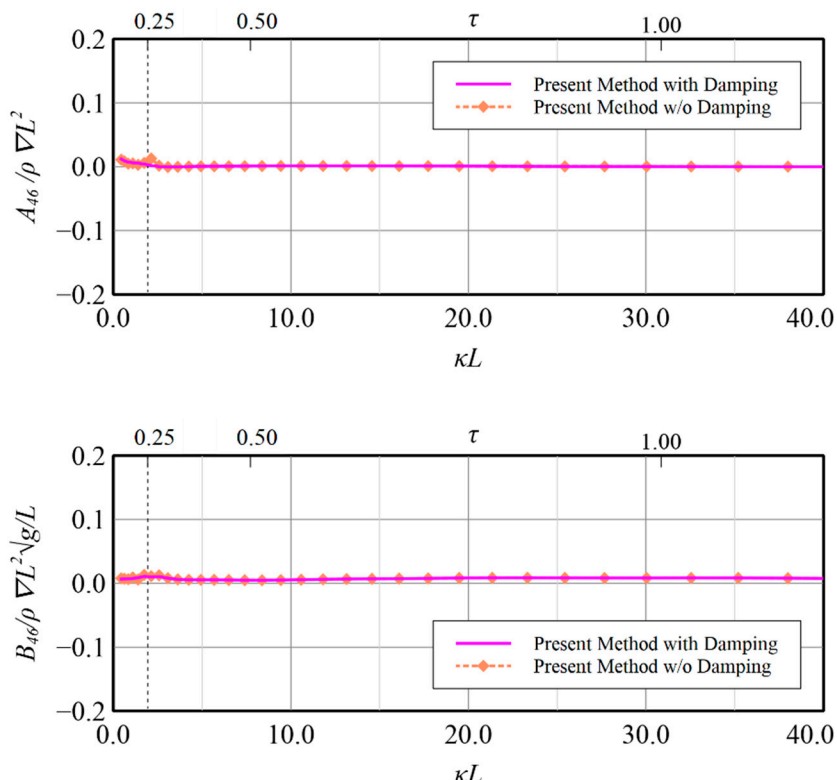

**Figure 26.** Yaw-induced-roll added mass and damping coefficient at $F_r = 0.18$.

## 5. Conclusions

The radiation of generated waves, which caused by an oscillating and translating body on a free surface, must be modeled accurately in the Rankine source model to avoid the flow field from the influence of wave reflection around the outward boundary. Specifically, under the overcritical condition ($\tau > 0.25$), the generated waves propagate backward behind the body, whereas under the undercritical condition ($\tau < 0.25$), the generated waves (especially the circular waves) scatter ahead of the body; this different feature indicates that proper numerical means must be chosen when modeling flow on a free surface.

In this paper, a frequency-domain Rankine source method based on a biquadratic B-spline scheme is presented; it involves an improved radiation mechanism and solves the flow problem with no restriction to $\tau$, making it have broader applicability and practical usefulness. Regarding the improved radiation mechanism in the present method, the simplified Seto's radiation boundary conditions are employed upstream and the introduction of spatially varying Rayleigh artificial damping over the free surface is incorporated. Both means are explicit and efficient and do not require control surface allocation in the boundary integral equation.

In evaluations conducted for cases with an oscillating and translating submerged source singularity, the proposed method is compared against the analytic solution obtained from the translating-pulsating-source Green function. The improved radiation boundary mechanism derives accurate wave radiation modelling, especially for the undercritical condition. The hydrodynamic forces induced by the forced oscillating movement of a translating bulker vessel are solved by the present method, and the obtained results, based on with and without introducing Rayleigh artificial damping, have good agreement in comparison with experimental data and public numerical prediction. Further investigation on the results with and without damping illustrates that proposed improved radiation mechanism, i.e., introducing Rayleigh artificial damping in additional to the simplified Seto's radiation boundary condition, can provide wider feasibility in various flow conditions.

In conclusion, the present B-spline Rankine source method, incorporating simplified Seto's radiation boundary condition and Rayleigh artificial damping, provides accurate flow

solutions under various $\tau$ conditions, and affords flexibility and computational efficiency. It can thus be useful to naval engineering in offshore and deep-water applications.

**Author Contributions:** Conceptualization, C.-H.W. and M.-C.F.; methodology, C.-H.W.; validation, C.-H.W.; writing—original draft preparation, C.-H.W.; writing—review and editing, M.-C.F.; supervision, M.-C.F. All authors have read and agreed to the published version of the manuscript.

**Funding:** This research received no external funding.

**Data Availability Statement:** The experimental data used for validation in the study can be found in: https://doi.org/10.2534/jjasnaoe.24.129 (accessed on 10 September 2022).

**Acknowledgments:** The authors want to thank Masashi Kashiwagi for kindly offering the related data about ship model geometry of RIOS bulker which is very useful for the present study.

**Conflicts of Interest:** The authors declare no conflict of interest.

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
