# Peer review of "Prediction of the Hydrodynamic Forces for a Ship Oscillating in Calm Water by an Improved Higher Order Rankine Panel Method"

_jmse, doi:10.3390/jmse10101337_

Round 1
Reviewer 1 Report
Title: Prediction Of The Hydrodynamic Forces For A Ship Oscillating In Calm Water By An Improved Higher Order Rankine Panel Method
In this manuscript, the authors have presented a Rankine source model based to analyze added mass/damping coefficient and validated with experimental results as well as another numerical method. In evaluation studies, the proposed method is successfully used to solve for the flow field generated by a single submerged disturbance source under critical and overcritical conditions for checking its effectiveness of proposed scheme. The subject is of importance and interest. The article is very well written and the authors put effort into writing the manuscript.
There are the following comments which are needed to be addressed before considering it for publication.
1. Introduction Section, more papers need to be reviewed, for instance,
Lyu W. and el Moctar O. (2017) Numerical and Experimental Investigations of Wave-induced Second Order Hydrodynamic Loads, Ocean Engineering 131:197–212.
2. Figure 18, the difference between present method and experimental results should be more specifically described. The experimental results are not shown in the short kL.
Reviewer 2 Report
The authors start by mentioning Green function methods as a justification of their present work on Rankine methods. However, their coverage of the Green function methods is rather narrow and does not cover various types of less conventional Green function formulations such as ones dealing with the acceleration potential of the transient Green function, to mention only some examples.
http://dx.doi.org/10.1016/j.enganabound.2017.09.005
https://doi.org/10.1016/j.oceaneng.2011.02.002
Although a full literature review is not expected here, more diversified references would be appropriate
The authors make the presentation of the method and successfully compare the results with the experimental data of the hydrodynamic coefficients of a bulker.
While the work is validated for the present conditions of deep water, it would be interesting to understand whether this might be applicable (or what would be necessary to make it applicable) to shallow water, where data of hydrodynamics coefficients are available for example in:
https://doi.org/10.1016/j.oceaneng.2010.03.008
The manuscript is generally well written but, on several occasions, there is no space between text and references or text and symbols.
